

# Technical note: Displacement variance of a solute particle in heterogeneous confined aquifers with random aquifer thickness fields

**Ching-Min Chang[1], Chuen-Fa Ni[1], Chi-Ping Lin[2], and I-Hsien Lee[2]**

[1]Graduate Institute of Applied Geology, National Central University, Taoyuan, Taiwan

[2]Center for Environmental Studies, National Central University, Taoyuan, Taiwan

**Correspondence:** Chuen-Fa Ni (nichuenfa@geo.ncu.edu.tw)



**Abstract.** In this work, the problem of regional-scale transport of inert solutes in
heterogeneous confined aquifers of variable thickness is analyzed in a stochastic
framework. A general stochastic methodology for deriving the variance of the
displacement of a solute particle is given based on the two-dimensional
depth-averaged solute mass conservation equation and the Fokker-Planck equation.
The variability in solute displacement is attributed to the variability in hydraulic
conductivity and aquifer thickness. Explicit results for the solute displacement
variance in the mean flow direction are obtained assuming that the fluctuations in log
hydraulic conductivity and log thickness of the confined aquifer are second-order
stationary processes. The results show that variation in hydraulic conductivity and
aquifer thickness can lead to nonstationarity in the covariance of flow velocity,
making longitudinal macrodispersion anomalous and increasing linearly with
travel time at large distances.
**1 Introduction**
It is widely accepted that the variability of solute movement in heterogeneous
aquifers is controlled primarily by the spatial variability of groundwater flow
fields (e.g., Dagan, 1989; Gelhar, 1993; Rubin, 2003). Much work on the



stochastic analysis of solute transport in heterogeneous porous formations has
focused on relating the spatial variability of the hydraulic conductivity field to
that of the flow velocity field, and thus to the spatial variability of the
displacement of a solute particle. However, natural aquifers at regional scales often
exhibit nonuniform aquifer thickness (e.g., Masterson et al., 2013; Zamrsky et al.,
2018; DeSimone et al., 2020), and spatial variability in the aquifer thickness field has
also been shown to have an important influence on flow field variability (e.g.,
Hantush, 1962; Cuello and Guarracino, 2020; Chang et al., 2021). Thus, the
underlying motivation for this work is to provide an analytical stochastic method for
improved quantification of the variability of solution displacement at the regional
scale in heterogeneous aquifers under more realistic field conditions, i.e., taking into
account the effects not only of the spatial variation of the hydraulic conductivity field
but also of the thickness field of the confined aquifer.

At a regional scale, the lateral extent of the confined aquifer is much greater than

the thickness of the formation. Therefore, it is more practical to view the flow and
solute transport processes in confined aquifers at the regional scale as essentially
two-dimensional, areal processes. In the traditional approach to the essentially
horizontal flow, the stochastic description of flow and solute transport processes is
related to the stochastic properties of transmissivity (e.g., Dagan, 1982; 1984), where

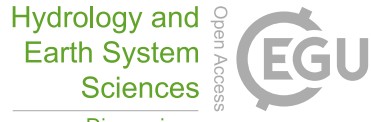

the transmissivity is the line integration of hydraulic conductivity over the depth of
the formation at a given point. However, in reality, transmissivity measurements from
field tests give a value of integrated hydraulic conductivity over a larger volume than
the range used for the line integration of hydraulic conductivity at a single point. This
means that the field tests performed for the transmissivity measurements include more
of the heterogeneity in the formation than that encountered over the depth of the
formation at a single point. This would result in a reduction in the variance of
transmissivity and an overestimation of the integral scale of transmissivity compared
to values predicted from the line integration of hydraulic conductivity. Consequently,
using the stochastic properties of transmissity may not provide an accurate
interpretation of solute movement at a regional scale.
Rather than using the stochastic properties of transmissity, this work uses the
stochastic properties of hydraulic conductivity and thickness of the confined
aquifer to interpret the variability of solute movement at a regional scale using a
hydraulic approach (or essentially horizontal flow approach) (Bear, 1979; Bear
and Cheng, 2010). That is, in this approach, the variability in solute movement is
due to variations in hydraulic conductivity and aquifer thickness. In the present
work, a general stochastic methodology is developed to describe the variability of
regional-scale solute transport, e.g., the variance of solute displacement, based on the





two-dimensional depth-averaged solute mass conservation equation and the
Fokker-Planck equation. Explicit results for the displacement variance in the mean
flow direction are obtained for the case where the random fields of log conductivity
and log thickness of the confined aquifer are second-order stationary. To our
knowledge, variation in hydraulic conductivity and aquifer thickness have not
previously been used as driving forces to quantify the variability of solute movement
in essentially horizontal flow fields. The stochastic theory presented here improves
quantification of the variance of the solute displacement in natural confined aquifers
of random thickness fields.

**2    Mathematical formulation of the problem**

Consider here the steady flow of a fluid carrying an inert solute through a
heterogeneous confined aquifer with variable thickness. When the dispersion tensor is
expressed in its three principal directions and these principal directions are used as
Cartesian coordinate axes, the equation for the transport of inert solutes through a rigid,
saturated porous medium is (e.g., de Marsily, 1986)
$$n\frac{\partial c}{\partial t} = \frac{\partial}{\partial x_i}\left[D_i\frac{\partial c}{\partial x_i} - cq_i\right] \qquad i = 1,2,3,$$    (1)
where $n$ is the porosity, $c$ is the solute concentration, and $D_i$ and $q_i$ are the dispersion





coefficient and the specific discharge in the $x_i$ direction, respectively. In the case where
the constituents are well mixed across the thickness of the aquifer (flow depth), it is
convenient to define the depth-averaged concentration in the $x_1$ and $x_2$ directions.

Integrating Eq. (1) with respect to $x_3$ over the vertical thickness of a confined

aquifer, $B(x_1, x_2)$, together with Leibniz's rule and no-slip condition for the dispersive
and diffusive fluxes at upper and lower boundaries of the confined aquifer, yields the
two-dimensional, depth-averaged equation for conservation of solute mass (e.g., Holly,
1975; Fischer et al., 1979)
$$\frac{\partial}{\partial t}[B\tilde{c}] = \frac{\partial}{\partial x_i}\left[\frac{\tilde{D}_i}{n}B\frac{\partial \tilde{c}}{\partial x_i}\right] - \frac{\partial}{\partial x_i}\left[\frac{\tilde{q}_i}{n}B\tilde{c}\right] \qquad i=1,2,$$    (2)
where   $\tilde{D}_i$ ,   $\tilde{c}$ ,  and   $\tilde{q}_i$   represent the depth-averaged dispersion coefficient,
depth-averaged   solute   concentration,   and   depth-averaged   specific   discharge,
respectively. Starting from the identity,
$$\frac{\tilde{D}_i}{n}B\frac{\partial \tilde{c}}{\partial x_i} = \frac{\partial}{\partial x_i}\left[\frac{\tilde{D}_i}{n}B\tilde{c}\right] - \tilde{c}\frac{\partial}{\partial x_i}\left[\frac{\tilde{D}_i}{n}B\right]$$
$$= \frac{\partial}{\partial x_i}\left[\frac{\tilde{D}_i}{n}B\tilde{c}\right] - B\tilde{c}\frac{1}{n}\frac{\partial \tilde{D}_i}{\partial x_i} - \frac{\tilde{D}_i}{n}B\tilde{c}\frac{\partial \ln B}{\partial x_i} \qquad i=1,2,$$    (3)
Eq. (2) can be rewritten as follows:
$$\frac{\partial}{\partial t}[B\tilde{c}] = \frac{\partial^2}{\partial x_i^2}\left[\frac{\tilde{D}_i}{n}B\tilde{c}\right] - \frac{\partial}{\partial x_i}\left[\left(\frac{1}{n}\frac{\partial \tilde{D}_i}{\partial x_i} + \frac{\tilde{D}_i}{n}\frac{\partial \ln B}{\partial x_i} + \frac{\tilde{q}_i}{n}\right)B\tilde{c}\right] \qquad i=1,2,$$    (4)
which corresponds to the form of the Fokker-Planck equation (e.g., Risken, 1989).

The concentration field associated with the solute particle can be written as

(Fischer et al., 1979; Dagan, 1989)



$$B\tilde{c}=\frac{M}{n}f(\boldsymbol{x};t,\boldsymbol{a},t_0),$$  (5)
where $M$ is the solute mass, $f(\boldsymbol{x};t,\boldsymbol{a},t_0)$ stands for the probability density function of the
particle displacement which originates at $\boldsymbol{x} = \boldsymbol{a}$ for $t = t_0$. Substituting Eq. (5) into Eq.
(4) gives
$$\frac{\partial}{\partial t}f(\boldsymbol{x};t)=\frac{\partial^2}{\partial x_i^2}[\frac{\tilde{D}_i}{n}f(\boldsymbol{x};t)]-\frac{\partial}{\partial x_i}[(\frac{1}{n}\frac{\partial \tilde{D}_i}{\partial x_i}+\frac{\tilde{D}_i}{n}\frac{\partial \ln B}{\partial x_i}+\frac{\tilde{q}_i}{n})f(\boldsymbol{x};t)]\qquad i=1,2,$$  (6)
which is known as the Fokker-Planck equation. Moreover, it can be shown that the
stochastic differential equation for the evolution of stochastic process (e.g., Van
Kampen, 1992; Jing et al., 2019)
$$\frac{dX_i}{dt}=\mu_i(\boldsymbol{X}(t))+\sigma_i(\boldsymbol{X}(t))\frac{dW}{dt}\qquad i=1,2,$$  (7)
where $\boldsymbol{X}(=(X_1,X_2))$ is the displacement, $\mu_i$ is the drift coefficient, $\sigma_i$ is the diffusion
coefficient, and $W$ denotes a Wiener process, is equivalent to the Fokker-Planck
equation (6) such that
$$\mu_i=\frac{1}{n}\frac{\partial}{\partial X_i}\tilde{D}_i(\boldsymbol{X})+\frac{1}{n}\tilde{D}_i(\boldsymbol{X})\frac{\partial}{\partial X_i}\ln B(\boldsymbol{X})+\frac{1}{n}\tilde{q}_i(\boldsymbol{X})\qquad i=1,2,$$  (8a)
$$\sigma_i^2=\frac{2}{n}\tilde{D}_i(\boldsymbol{X})\qquad i=1,2,$$  (8b)
This means,
$$\frac{dX_i}{dt}=[\frac{1}{n}\frac{\partial}{\partial X_i}\tilde{D}_i(\boldsymbol{X})+\frac{1}{n}\tilde{D}_i(\boldsymbol{X})\frac{\partial}{\partial X_i}\ln B(\boldsymbol{X})+\frac{1}{n}\tilde{q}_i(\boldsymbol{X})]+\sqrt{\frac{2}{n}\tilde{D}_i(\boldsymbol{X})}\frac{dW}{dt}\qquad i=1,2.$$  (9)
In this study, the fields (or processes) of hydraulic conductivity $K(x_1,x_2)$ and
thickness of the confined aquifer $B(x_1,x_2)$ are considered spatially random. It is also
assumed that the mean fluid flow is uniform and unidirectional in the $x_1$-direction, i.e.,





the gradient of the mean depth-averaged hydraulic head $\tilde{h}$ in the $x_1$-direction is
constant,
$$\frac{d<\tilde{h}>}{dx_1} = -J,$$    (10)
and zero in the $x_2$-direction, which is consistent with the result of the perturbation
approach from the solution of the differential equation for the depth-averaged
hydraulic head (Chang et al., 2021). Thus, $<X> = (<X_1>, 0)$ and the depth-averaged
dispersion coefficients in Eq. (9) become constant, $\tilde{D}_1 = D_L$ and $\tilde{D}_2 = D_T$.

By analogy with Butera and Tanda (1999), the expansion of Eq. (9) in Taylor

series around $<X>$ in the $x_1$-direction yields
$$\frac{dX_1}{dt} = \frac{1}{n}D_L\left[\frac{d\Phi(<X>)}{dX_1} + \frac{d^2\Phi(<X>)}{dX_1^2}X_1' + \frac{d\beta(<X>)}{dX_1}\right] + <\tilde{v}_1>(<X>) + v_1<X> + \sqrt{\frac{2}{n}D_L}\frac{dW}{dt},$$    (11)
where $X_1' = X_1-<X_1>$, $\Phi = <\ln B>$, $\beta = \ln B-<\ln B>$, $v_1 = \tilde{v}_1-<\tilde{v}_1>$, and $\tilde{v}_1 = \tilde{q}_1/n$. Note
that due to the assumption of uniform mean flow in the $x_1$-direction, the term
$$\frac{d<\tilde{v}_1>(<X>)}{dX_1}X_1',$$    (12)
has been removed from Eq. (11). Equation (11) reveals that
$$\frac{d<X_1>}{dt} = \frac{1}{n}D_L\frac{d\Phi(<X>)}{dX_1} + <\tilde{v}_1>(<X>),$$    (13)
$$\frac{dX_1'}{dt} - \frac{D_L}{n}\frac{d^2\Phi(<X>)}{dX_1^2}X_1' = \frac{D_L}{n}\frac{d\beta(<X>)}{dX_1} + v_1<X> + \sqrt{\frac{2}{n}D_L}\frac{dW}{dt}.$$    (14)
Equations (13) and (14) describe the mean and fluctuation, respectively, of the
displacement of the solute particles. By the solution of Eq. (14), the variance of the
solute displacement in the $x_1$-direction (the mean flow direction) can be evaluated in





the frame, (e.g., Dagan, 1984; 1989)
$X_{11}(t) = <X_1^{'}(t)X_1^{'}(t)>.$                                                (15)

This first-order approximation for representing the head perturbation, and hence

the solute displacement perturbation, should be applied to porous formations where
the standard deviation of the random fluctuations of the log conductivity is less than 1.
However, Zhang and Winter (1999) report in a Monte Carlo simulation study that it is
accurate for the solutions of the head moment for the value of the variance of the log
conductivity of up to 4.38. A similar finding from comparing moments of hydraulic
head with results of numerical Monte Carlo simulations is also reported in Guadagnini
and Neuman (1999) for highly heterogeneous media with a variance of log
conductivity from 2 to 4.

For the case of the aquifer thickness which is a slowly spatial varying process, the

term in Eq. (14), $d^2\Phi/dX_1^2$, may be neglected, and, consequently, Eq. (14) reduces to
$\dfrac{dX_1^{'}}{dt} = \dfrac{D_L}{n}\dfrac{d\beta(<X>)}{dX_1} + v_1<X> + \sqrt{\dfrac{2}{n}D_L}\dfrac{dW}{dt}.$          (16)
Equation (16) illustrates that the variability of the particle displacement is determined
by the gradient of the variation of the aquifer thickness fields, the variability of the
flow velocity and the local pore-scale dispersion. Note that when flowing through a
confined aquifer with variable thickness, the variability in flow velocity is influenced
by both the variation in log conductivity and log thickness fields (Chang et al., 2021).



Equations (15) and (16) form a basis of this study for the development of the
displacement variance in the mean flow direction.
Since the variance of solute displacement in the $x_1$-direction cannot be calculated
without knowing the statistics of the flow fields. Therefore, in the following section,
the statistics of the flow field are developed for the case where both the variations in
hydraulic conductivity and the thickness of the confined aquifer are considered to be
second-order stationary processes.

**3    Statistics of the flow fields**

Chang et al. (2021) develop the differential equations for the flow fields (Eqs. (6) and
(12) of Chang et al., 2021) in a confined aquifer with variable thickness based on a
hydraulic approach to flow in aquifers (Bear, 1979; Bear and Cheng, 2010). On this
basis, under the condition of steady-state flow, the equations for the depth-averaged
hydraulic head and the depth-averaged specific discharge about the mean, keeping
only first-order terms in the perturbations, take the following form
$$\frac{\partial^2 h}{\partial x_i^2} = J\left[\frac{\partial y}{\partial x_1} + 2\frac{\partial \beta}{\partial x_1}\right] \qquad i = 1,2,$$    (17)
$$q_i = \bar{q}\left[(y+\beta)\delta_{1i} - \frac{1}{J}\frac{\partial h}{\partial x_i}\right] \qquad i = 1,2,$$    (18)
where $h = \tilde{h} - <\tilde{h}>$, $y = \ln K - Y$, $Y = <\ln K>$, $\beta = \ln B - <\ln B>$, $q_i = \tilde{q}_i - <\tilde{q}_i>$, and





$\bar{q} =< \tilde{q}_1 >= e^y J$ . Equation (17) quantifies the change in depth-averaged head in
response to changes in hydraulic conductivity and aquifer thickness.

Due to the property of the linearity of the driving forces, Eq. (17) can

alternatively be divided into two parts as
$\dfrac{\partial^2 h_y}{\partial x_i^2} = J \dfrac{\partial y}{\partial x_1} \qquad i = 1,2,$ (19a)
$\dfrac{\partial^2 h_\beta}{\partial x_i^2} = 2J \dfrac{\partial \beta}{\partial x_1} \qquad i = 1,2,$ (19b)
where $h = h_y + h_\beta$. Equation (19) is a stochastic differential equation in which the
variation in log-hydraulic conductivity (or in log-aquifer thickness) appears as a
forcing term that produces the variations in depth-averaged head.

Matheron (1973) shows that if the random input process of the Poission equation

is second-order stationary, then the Poission equation has a first-order intrinsic random
function (1-IRF) as its solution. Since the processes $y$ and $\beta$ are second-order
stationary, it can be shown that the derivatives of the processes $y$ and $\beta$ with respect to
$x_1$ are also stationary. This means that Eq. (19) has a 1-IRF solution for $h_y$ and $h_\beta$ which
admits the Fourier-Stieltjes representation as follows:
$h_y(x_1, x_2) = J \displaystyle\int\limits_{-\infty}^{\infty}\int\limits_{-\infty}^{\infty} i R_1 \dfrac{1 - \exp[i(R_1 x_1 + R_2 x_2)] + i(R_1 x_1 + R_2 x_2)}{R_1^2 + R_2^2} dZ_y(R_1, R_2),$ (20a)
$h_\beta(x_1, x_2) = 2J \displaystyle\int\limits_{-\infty}^{\infty}\int\limits_{-\infty}^{\infty} i R_1 \dfrac{1 - \exp[i(R_1 x_1 + R_2 x_2)] + i(R_1 x_1 + R_2 x_2)}{R_1^2 + R_2^2} dZ_\beta(R_1, R_2).$ (20b)





where $R_1$ and $R_2$ are the components of the wave number vector $\boldsymbol{R}$ ($= (R_1, R_2)$), and $Z_y$
and $Z_\beta$ are complex-valued distributions with uncorrelated increments on wave
number space. Note that a 1-IRF is the second integral of a zero-mean spatial random
function (Chile`s and Delfiner, 1999).

It follows from Eq. (18) that if processes $y$ and $\beta$ are statistically independent, the

covariance function for the depth-averaged flow velocity process can be evaluated
with the spectral representation as follows:
$$\frac{<v_i(\boldsymbol{\xi})v_j(\boldsymbol{\zeta})>}{V^2} = [C_{yy}(\boldsymbol{\xi},\boldsymbol{\zeta}) + C_{\beta\beta}(\boldsymbol{\xi},\boldsymbol{\zeta})]\delta_{1i}\delta_{1j} - \frac{1}{J}\frac{\partial}{\partial\zeta_j}[C_{yh_y}(\boldsymbol{\xi},\boldsymbol{\zeta}) + C_{\beta h_\beta}(\boldsymbol{\xi},\boldsymbol{\zeta})]\delta_{1i}$$
$$-\frac{1}{J}\frac{\partial}{\partial\xi_i}[C_{yh_y}(\boldsymbol{\zeta},\boldsymbol{\xi}) + C_{\beta h_\beta}(\boldsymbol{\zeta},\boldsymbol{\xi})]\delta_{1j} - \frac{1}{J^2}\frac{\partial\gamma_h(\boldsymbol{\xi},\boldsymbol{\zeta})}{\partial\xi_i\partial\zeta_j}, \qquad (21)$$
where $V = \overline{q}/n = e^Y J/n$, $\boldsymbol{\xi} = (\xi_1,\xi_2)$, $\boldsymbol{\zeta} = (\zeta_1,\zeta_2)$, $C_{yy}$ and $C_{\beta\beta}$ are the $\ln K$ and $\ln B$
covariance functions, respectively, $C_{yh_y}$ is the covariance of $\ln K$ process with the head
process, $C_{\beta h_\beta}$ is the covariance of $\ln B$ process with the head process, and $\gamma_h$ is the
semivariogram of the head process, defined as
$$\gamma_h(\boldsymbol{\xi},\boldsymbol{\zeta}) = \gamma_{h_y}(\boldsymbol{\xi},\boldsymbol{\zeta}) + \gamma_{h_\beta}(\boldsymbol{\xi},\boldsymbol{\zeta}) = \frac{1}{2}\left\{<[h_y(\boldsymbol{\xi}) - h_y(\boldsymbol{\zeta})]^2> + <[h_\beta(\boldsymbol{\xi}) - h_\beta(\boldsymbol{\zeta})]^2>\right\}. \quad (22)$$

**4    Results and discussion**

To determine the covariance of flow velocity, and thus the variance of solute
displacement, it is assumed that the hydraulic conductivity and the thickness of the





aquifer fields are lognormally distributed and characterized by the isotropic
exponential covariance, i.e. (e.g., Dagan, 1984; Gelhar, 1993; Bailey and Baù, 2012)
$$C_{yy}(\boldsymbol{\xi},\boldsymbol{\zeta})=\sigma_y^2\exp[-\frac{|\boldsymbol{\xi}-\boldsymbol{\zeta}|}{\lambda_y}],\qquad(23a)$$
$$C_{\beta\beta}(\boldsymbol{\xi},\boldsymbol{\zeta})=\sigma_\beta^2\exp[-\frac{|\boldsymbol{\xi}-\boldsymbol{\zeta}|}{\lambda_\beta}],\qquad(23b)$$
where $\sigma_y^2$ and $\sigma_\beta^2$ are the variances of $y$ and $\beta$, respectively, $\lambda_y$ and $\lambda_\beta$ are the integral
scales of $\ln K$ and $\ln B$ fields, respectively. The corresponding spectra, which result
from the inverse Fourier transform of Eq. (23), are as follows:
$$S_{yy}(R_1,R_2)=\frac{\sigma_y^2}{2\pi}\frac{\lambda_y^2}{[1+\lambda_y^2(R_1^2+R_2^2)]^{3/2}},\qquad(24a)$$
$$S_{\beta\beta}(R_1,R_2)=\frac{\sigma_\beta^2}{2\pi}\frac{\lambda_\beta^2}{[1+\lambda_\beta^2(R_1^2+R_2^2)]^{3/2}}.\qquad(24b)$$

**4.1   Covariance of flow velocity in the $x_1$-direction**

The stationarity of the $\ln K$ process allows the Fourier-Stieltjes representations (e.g.,
Lumley and Panofsky, 1964)
$$y(x_1,x_2)=\int_{-\infty}^{\infty}\int_{-\infty}^{\infty}\exp[i(R_1x_1+R_2x_2)]dZ_y(R_1,R_2).\qquad(25)$$
Using this and Eqs. (20a) and (24a), the covariance of $\ln K$ process with the head
process $C_{yh}$ in Eq. (21) is given as
$$C_{yh_y}(\boldsymbol{\xi},\boldsymbol{\zeta})=<y(\boldsymbol{\xi})h_y(\boldsymbol{\zeta})>$$





$$= -J \int\limits_{-\infty}^{\infty} \int\limits_{-\infty}^{\infty} i \frac{R_1}{R_1^2 + R_2^2} \exp[i(R_1\xi_1 + R_2\xi_2)] \{1 - \exp[-i(R_1\zeta_1 + R_2\zeta_2)] - i(R_1\zeta_1 + R_2\zeta_2)\}$$

$$\times \frac{\sigma_y^2}{2\pi} \frac{\lambda_y^2}{[1 + \lambda_y^2(R_1^2 + R_2^2)]^{3/2}} dR_1 dR_2$$

$$= \sigma_y^2 \lambda_y J \left[ \Theta_1(\frac{\xi_1}{\lambda_y}, \frac{\xi_2}{\lambda_y}) - \frac{\zeta_1}{\lambda_y} \Theta_2(\frac{\xi_1}{\lambda_y}, \frac{\xi_2}{\lambda_y}) + \frac{\zeta_2}{\lambda_y} \Theta_3(\frac{\xi_1}{\lambda_y}, \frac{\xi_2}{\lambda_y}) - \Theta_1(\frac{\rho_1}{\lambda_y}, \frac{\rho_2}{\lambda_y}) \right],$$ (26)

where $\rho_1 = \xi_1 - \zeta_1$, $\rho_2 = \xi_2 - \zeta_2$, and the description for the functions $\Theta_1$-$\Theta_3$, respectively,
can be found in Appendix A. Similarly, the closed-form expression for the covariance
of $\ln B$ process with the head process $C_{\beta h_\beta}$ in Eq. (21) can be obtained using Eqs. (20b),
(24b), and the Fourier-Stieltjes representations for the stationary $\ln B$ process
$$\beta(x_1, x_2) = \int\limits_{-\infty}^{\infty} \int\limits_{-\infty}^{\infty} \exp[i(R_1 x_1 + R_2 x_2)] dZ_\beta(R_1, R_2),$$ (27)

which is in the form
$$C_{\beta h_\beta}(\boldsymbol{\xi}, \boldsymbol{\zeta}) = <\beta(\boldsymbol{\xi}) h_\beta(\boldsymbol{\zeta})>$$

$$= 2\sigma_y^2 \lambda_\beta J \left[ \Theta_1(\frac{\xi_1}{\lambda_\beta}, \frac{\xi_2}{\lambda_\beta}) - \frac{\zeta_1}{\lambda_\beta} \Theta_2(\frac{\xi_1}{\lambda_\beta}, \frac{\xi_2}{\lambda_\beta}) + \frac{\zeta_2}{\lambda_\beta} \Theta_3(\frac{\xi_1}{\lambda_\beta}, \frac{\xi_2}{\lambda_\beta}) - \Theta_1(\frac{\rho_1}{\lambda_\beta}, \frac{\rho_2}{\lambda_\beta}) \right].$$ (28)

Substituting Eq. (20) into Eq. (22), it is found that the semivariogram of the head
process has the following form
$$\gamma_{h_y}(\boldsymbol{\xi}, \boldsymbol{\zeta}) = \frac{1}{2} \sigma_y^2 \lambda_y^2 J^2 \left\{ \frac{3}{8} \frac{\rho_1^2}{\lambda_y^2} + \frac{1}{8} \frac{\rho_2^2}{\lambda_y^2} + \Psi_1(\frac{\rho_1}{\lambda_y}, \frac{\rho_2}{\lambda_y}) + \frac{\rho_1}{\lambda_y} [-\frac{\xi_1}{\lambda_y} \Psi_2(\frac{\xi_1}{\lambda_y}, \frac{\xi_2}{\lambda_y}) + \frac{\zeta_1}{\lambda_y} \Psi_2(\frac{\zeta_1}{\lambda_y}, \frac{\zeta_2}{\lambda_y})] \right.$$

$$+ \frac{\rho_2}{\lambda_y} [\frac{\xi_2}{\lambda_y} \Psi_3(\frac{\xi_1}{\lambda_y}, \frac{\xi_2}{\lambda_y}) - \frac{\zeta_2}{\lambda_y} \Psi_3(\frac{\zeta_1}{\lambda_y}, \frac{\zeta_2}{\lambda_y})] \right\},$$ (29a)

$$\gamma_{h_\beta}(\boldsymbol{\xi}, \boldsymbol{\zeta}) = 2\sigma_\beta^2 \lambda_\beta^2 J^2 \left\{ \frac{3}{8} \frac{\rho_1^2}{\lambda_\beta^2} + \frac{1}{8} \frac{\rho_2^2}{\lambda_\beta^2} + \Psi_1(\frac{\rho_1}{\lambda_\beta}, \frac{\rho_2}{\lambda_\beta}) + \frac{\rho_1}{\lambda_\beta} [-\frac{\xi_1}{\lambda_\beta} \Psi_2(\frac{\xi_1}{\lambda_\beta}, \frac{\xi_2}{\lambda_\beta}) + \frac{\zeta_1}{\lambda_\beta} \Psi_2(\frac{\zeta_1}{\lambda_\beta}, \frac{\zeta_2}{\lambda_\beta})] \right.$$

$$+ \frac{\rho_2}{\lambda_\beta} [\frac{\xi_2}{\lambda_\beta} \Psi_3(\frac{\xi_1}{\lambda_\beta}, \frac{\xi_2}{\lambda_\beta}) - \frac{\zeta_2}{\lambda_\beta} \Psi_3(\frac{\zeta_1}{\lambda_\beta}, \frac{\zeta_2}{\lambda_\beta})] \right\},$$ (29b)



where $\rho_1 = \xi_1\text{-}\zeta_1$, $\rho_2 = \xi_2\text{-}\zeta_2$, and the description for the functions $\Psi_1$-$\Psi_3$, respectively,
can be found in Appendix B.

In the case of statistically nonhomogeneous random fields, the structure of

variability can be characterized by considering the semivariogram of a random field. If
the semivariogram depends only on the separation, the random field is said to have
stationary increments. The semivariogram in Eq. (29) clearly depends on the spatial
location, which means that the processes of depth-averaged hydraulic head are
nonstationary.

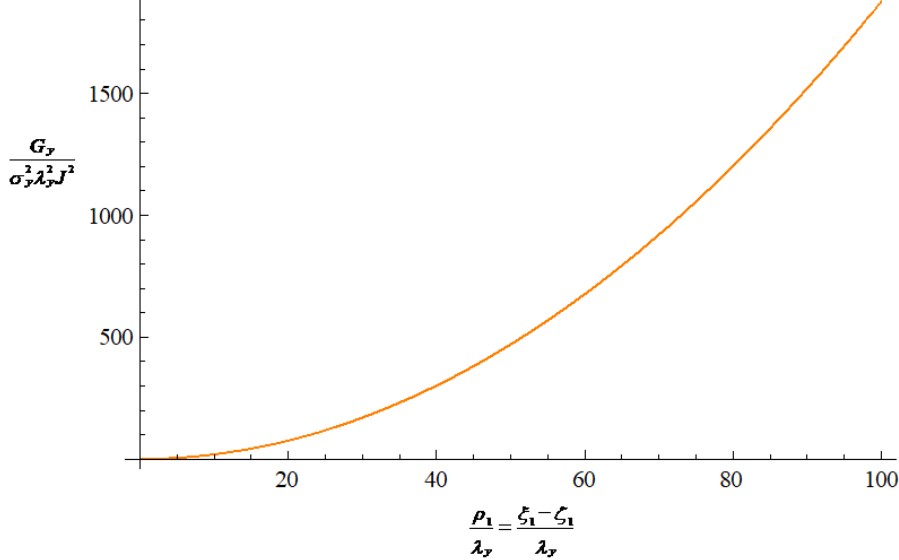


**Figure 1.** The stationary parts of the semivariogram of the head field, reflecting the
effect of variation in the hydraulic conductivity fields, as a function of the separation
distance in the mean flow direction, where $G_y$ is the sum of the first three terms on the
right-hand side of Eq. (29a).
Figure 1 shows graphically the behavior of the stationary parts of the
semivariogram (namely, the sum of the first three terms on the right-hand side of Eq.
(29a)) as a function of the separation distance in the $x_1$-direction (mean flow direction).
The semivariogram of the head field, reflecting the effect of variation in the hydraulic
conductivity fields, shows an unlimited increase, as shown in Fig. 1. The unbounded
head semivariogram suggests that there is no head covariance function (or the
hydraulic head field with infinite variance). In this case, the use of the semivariogram
is an appropriate way to measure the variability of the head variation. Similar
conclusions can be drawn from Fig. 2, a graphical representation of the stationary
parts of the semivariogram of the head field in Eq. (29b) in the mean flow direction,
which reflects the effect of the variation of the aquifer thickness fields.

At this point, the covariance function for the depth-averaged velocity process in

Eq. (21) can now be determined in conjunction with Eqs. (23), (26), (28), and (29).
For example, the covariance of flow velocity for the separation along the mean flow
direction is explicitly determined as follows:
$<v_1(\xi_1,\xi_2)v_1(\zeta_1,\zeta_2=\xi_2)>=<v_{y_1}(\xi_1,\xi_2)v_{y_1}(\zeta_1,\xi_2)>+<v_{\beta_1}(\xi_1,\xi_2)v_{\beta_1}(\zeta_1,\xi_2)>,$    (30a)
where
$\dfrac{<v_{y_1}(\xi_1,\xi_2)v_{y_1}(\zeta_1,\xi_2)>}{V^2}=\sigma_y^2\{\dfrac{3}{8}+\exp(-\dfrac{\rho}{\lambda_y})-[2\Xi_1(\dfrac{\rho_1}{\lambda_y},0)-\Xi_1(\dfrac{\xi_1}{\lambda_y},\dfrac{\xi_2}{\lambda_y})-\Xi_1(\dfrac{\zeta_1}{\lambda_y},\dfrac{\xi_2}{\lambda_y})]$
$+[\Xi_2(\dfrac{\rho_1}{\lambda_y},0)-\Xi_2(\dfrac{\xi_1}{\lambda_y},\dfrac{\xi_2}{\lambda_y})-\Xi_2(\dfrac{\zeta_1}{\lambda_y},\dfrac{\xi_2}{\lambda_y})]\},$    (30b)



$$\frac{<v_{\beta_1}(\xi_1,\xi_2)v_{\beta_1}(\zeta_1,\xi_2)>}{V^2}=\sigma_\beta^2\{\frac{3}{2}+\exp(-\frac{\rho}{\lambda_\beta})-2[2\varXi_1(\frac{\rho_1}{\lambda_\beta},0)-\varXi_1(\frac{\xi_1}{\lambda_\beta},\frac{\xi_2}{\lambda_\beta})-\varXi_1(\frac{\zeta_1}{\lambda_\beta},\frac{\xi_2}{\lambda_\beta})]$$

$$+4[\varXi_2(\frac{\rho_1}{\lambda_\beta},0)-\varXi_2(\frac{\xi_1}{\lambda_\beta},\frac{\xi_2}{\lambda_\beta})-\varXi_2(\frac{\zeta_1}{\lambda_\beta},\frac{\xi_2}{\lambda_\beta})]\},\qquad (30c)$$

$\rho=(\rho_1^2+\rho_2^2)^{1/2}$ and expressions for $\varXi_1$ and $\varXi_2$ are given, respectively, in the Appendix C.
This should be used to compute the variance of solute displacement in the mean flow
direction. The nonstationarity of the velocity covariance in Eq. (30) is evident in the
dependence on spatial location, which is caused by nonstationarity in the hydraulic
head processes.

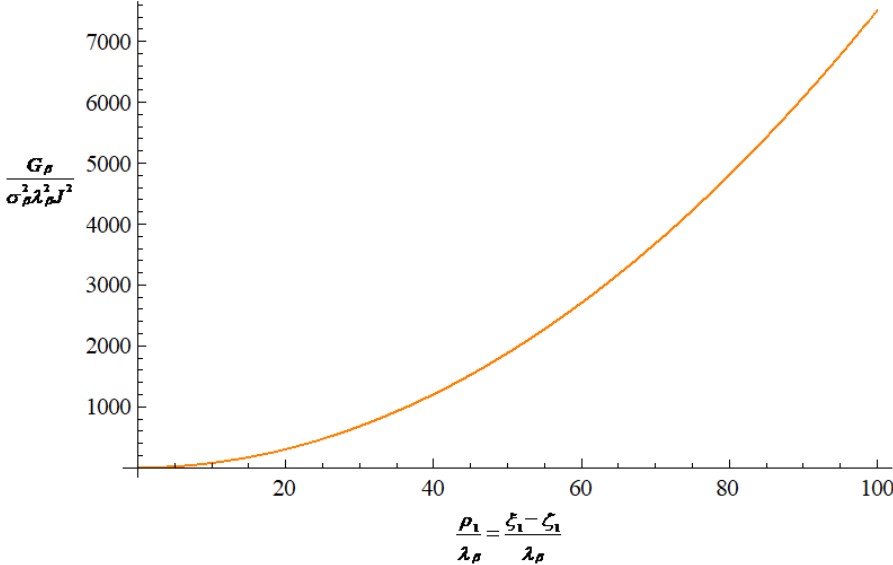


**Figure 2.** The stationary parts of the semivariogram of the head field, reflecting the
effect of the variation of the aquifer thickness fields, as a function of the separation
distance in the mean flow direction, where $G_\beta$ is the sum of the first three terms on the
right-hand side of Eq. (29b).
In the limit of $\zeta_1\rightarrow\xi_1$, Eq. (30) approaches to the velocity variances in the mean





flow direction as
$\sigma_v^2 = \sigma_{v_y}^2 (\xi_1, \xi_2) + \sigma_{v_\beta}^2 (\xi_1, \xi_2),$            (31a)
where
$\dfrac{\sigma_{v_y}^2}{V^2 \sigma_y^2} = \dfrac{1}{4}\dfrac{1}{\xi^8} \Delta_1(\dfrac{\xi_1}{\lambda_y}, \dfrac{\xi_2}{\lambda_y}) + \dfrac{1}{4}\dfrac{1}{\xi^9} \exp[-\dfrac{\xi}{\lambda_y}] \Delta_2(\dfrac{\xi_1}{\lambda_y}, \dfrac{\xi_2}{\lambda_y}),$            (31b)
$\dfrac{\sigma_{v_\beta}^2}{V^2 \sigma_\beta^2} = \dfrac{1}{2}\dfrac{1}{\xi^8} \Delta_3(\dfrac{\xi_1}{\lambda_\beta}, \dfrac{\xi_2}{\lambda_y}) + \dfrac{1}{2}\dfrac{1}{\xi^9} \exp[-\dfrac{\xi}{\lambda_\beta}] \Delta_4(\dfrac{\xi_1}{\lambda_\beta}, \dfrac{\xi_2}{\lambda_\beta}).$            (31c)
where $\xi = (\xi_1^2 + \xi_2^2)^{1/2}$ and expressions for $\Delta_1$-$\Delta_4$ are given, respectively, in the Appendix
D. From equation (31), it can be seen that the variance of the flow velocity is
positively correlated with the variances of the log-hydraulic conductivity and
log-aquifer thickness. This means that the variability of the flow velocity field
increases with the variability of the hydraulic conductivity and aquifer thickness fields.

**4.2   Variance of the solute displacement in the mean flow direction**

The stochastic Eq. (16) is more complex because the first term appears on the
right-hand side of Eq. (16). In general, it is not possible to explicitly derive the
relationships between the variance of solute displacement and that of the flow field
using the solution of Eq. (16). To take advantage of the closed form, this study
considers the case where the local dispersivity is very small compared to the integral



scales for the lnK and lnB processes, so that the solute dispersion is mainly caused by
the spatial variability of hydraulic conductivity and thickness of confined aquifer.
That is, solute dispersion occurs in situations where advection dominates and solute
particles do not transfer across streamlines. There are numerous studies in the
literature on solute transport under advection-dominated conditions, e.g., Dagan
(1984), Rubin and Bellin (1994), Butera et al. (2009), Cvetkovic (2016), Ciriello and
Barros (2020), etc.
In advection-dominated situations, the local dispersion coefficient $D_L$ in Eq. (16)
can be set to zero and Eq. (16) is simplified to
$$\frac{dX_1^{'}}{dt} = v_1(<X>),\qquad(32)$$
which gives the solution
$$X_1^{'}(t) = \int_0^t v_1(vS, 0)dS .\qquad(33)$$
This implies that the displacement variance can be expressed in terms of the flow
velocity covariance through the double integral as
$$X_{11}(t) = \int_0^t \int_0^t < v_1(vS_1, 0)v_1(vS_2, 0) > dS_1 dS_2 .\qquad(34)$$

**4.2.1      Nonstationary flow fields**






Substituting Eq. (30) into Eq. (34) and integrating it with $\xi_2 = 0$ yields the following
expression for the variance of longitudinal solute displacement as
$$X_{11}(t) = X_{11_y}(t) + X_{11_\beta}(t),$$ (35a)
where
$$\frac{X_{11_y}(t)}{\sigma_y^2 \lambda_y^2} = \frac{5}{2} - 3\gamma - \frac{9}{\Gamma^2} + 2\Gamma + \frac{3}{8}\Gamma^2 + 3Ei(-\Gamma) - 3\ln(\Gamma) + e^{-\Gamma}\left(2 + \frac{9}{\Gamma^2} + \frac{9}{\Gamma}\right),$$ (35b)
$$\frac{X_{11_\beta}(t)}{\sigma_\beta^2 \lambda_\beta^2} = 4 - 4\gamma - \frac{36}{\vartheta^2} + 2\vartheta + \frac{3}{2}\vartheta^2 + 4Ei(-\vartheta) - 4\ln(\vartheta) + 2e^{-\vartheta}\left(7 + 2\vartheta + \frac{18}{\vartheta^2} + \frac{18}{\vartheta}\right),$$ (35c)
$\Gamma = Vt/\lambda_y$, and $\vartheta = Vt/\lambda_\beta$.

**4.2.2    Stationary flow fields**

Gutjahr and Gelhar (1981) show that the Poission equation in an unbounded porous
medium such as equation (19a) also has a zero-order intrinsic random function (0-IRF)
as its solution when the input random process has a finite variance. That is, Eqs. (19a)
and (19b) with stationary processes $y$ and $\beta$ admit the solutions of the form
$$h_y(x_1, x_2) = J \int_{-\infty}^{\infty}\int_{-\infty}^{\infty} iR_1 \frac{1 - \exp[i(R_1 x_1 + R_2 x_2)]}{R_1^2 + R_2^2} dZ_y(R_1, R_2),$$ (36a)
$$h_\beta(x_1, x_2) = 2J \int_{-\infty}^{\infty}\int_{-\infty}^{\infty} iR_1 \frac{1 - \exp[i(R_1 x_1 + R_2 x_2)]}{R_1^2 + R_2^2} dZ_\beta(R_1, R_2).$$ (36b)





Using a similar methodology as above and based on Eq. (36), one would arrive at
the following results

$$C_{yh_y}(\boldsymbol{\xi},\boldsymbol{\zeta})=\sigma_y^2\lambda_y J\left[\Theta_1(\frac{\xi_1}{\lambda_y},\frac{\xi_2}{\lambda_y})-\Theta_1(\frac{\rho_1}{\lambda_y},\frac{\rho_2}{\lambda_y})\right],\qquad(37a)$$

$$C_{yh_y}(\boldsymbol{\xi},\boldsymbol{\zeta})=\sigma_y^2\lambda_y J\left[\Theta_1(\frac{\xi_1}{\lambda_y},\frac{\xi_2}{\lambda_y})-\Theta_1(\frac{\rho_1}{\lambda_y},\frac{\rho_2}{\lambda_y})\right],\qquad(37b)$$

$$\gamma_{h_y}(\boldsymbol{\xi},\boldsymbol{\zeta})=\frac{1}{2}\sigma_y^2\lambda_y^2 J^2\Psi_1(\frac{\rho_1}{\lambda_y},\frac{\rho_2}{\lambda_y}),\qquad(38a)$$

$$\gamma_{h_\beta}(\boldsymbol{\xi},\boldsymbol{\zeta})=2\sigma_\beta^2\lambda_\beta^2 J^2\Psi_1(\frac{\rho_1}{\lambda_\beta},\frac{\rho_2}{\lambda_\beta}),\qquad(38b)$$

from which it follows that in the mean flow direction,

$$<v_1(\xi_1,\xi_2)v_1(\zeta_1,\zeta_2=\xi_2)>=<v_{y_1}(\xi_1,\xi_2)v_{y_1}(\zeta_1,\xi_2)>+<v_{\beta_1}(\xi_1,\xi_2)v_{\beta_1}(\zeta_1,\xi_2)>,\qquad(39a)$$

where

$$\frac{<v_{y_1}(\xi_1,\xi_2)v_{y_1}(\zeta_1,\xi_2)>}{V^2}=\sigma_y^2\left[\frac{3}{2}\left(-\frac{6}{\varphi^4}+\frac{1}{\varphi^2}\right)+3e^{-\varphi}\left(\frac{3}{\varphi^4}+\frac{3}{\varphi^3}+\frac{1}{\varphi^2}\right)\right],\qquad(39b)$$

$$\frac{<v_{\beta_1}(\xi_1,\xi_2)v_{\beta_1}(\zeta_1,\xi_2)>}{V^2}=\sigma_\beta^2\left[-2\left(\frac{18}{\upsilon^4}+\frac{1}{\upsilon^2}\right)+e^{-\varphi}\left(1+\frac{36}{\upsilon^4}+\frac{36}{\upsilon^3}+\frac{16}{\upsilon^2}+\frac{4}{\upsilon}\right)\right],\qquad(39c)$$

$\varphi = (\xi_1\text{-}\zeta_1)/\lambda_y$ and $\upsilon = (\xi_1\text{-}\zeta_1)/\lambda_\beta$. Finally, the variance of solute displacement in the
mean flow direction is obtained from Eq. (34) by applying Eq. (39):

$$X_{11}(t)=X_{11_y}(t)+X_{11_\beta}(t),\qquad(40a)$$

where

$$\frac{X_{11_y}(t)}{\sigma_y^2\lambda_y^2}=\frac{3}{2}-3\gamma+2\Gamma-\frac{3}{\Gamma^2}+3Ei(-\Gamma)-3\ln(\Gamma)+3e^{-\Gamma}\left(\frac{1}{\Gamma^2}+\frac{1}{\Gamma}\right),\qquad(40b)$$

$$\frac{X_{11_\beta}(t)}{\sigma_\beta^2\lambda_\beta^2}=4-4\gamma-\frac{12}{\vartheta^2}+2\vartheta+4Ei(-\vartheta)-4\ln(\vartheta)+2e^{-\vartheta}\left(1+\frac{6}{\vartheta^2}+\frac{6}{\vartheta}\right).\qquad(40c)$$

Equation (40b) is equivalent to the solution of Dagan (1982; 1984) using the Green
function approach, where the variance and integral scale of the log conductivity fields


in Eq. (40b) are replaced by the variance and integral scale of the log transmissivity
fields.

A comparison of the prediction of the solute longitudinal displacement variance

in Eq. (35b) in nonstationary flow fields with the prediction in Eq. (40b) in stationary
flow fields is shown graphically in Fig. 3. The variance of the longitudinal
displacement in response to the change in the hydraulic conductivity grows
monotonically with travel time. It can also be seen that the difference in displacement
variance caused by the nonstationary and stationary flow fields increases with travel
time, which means that the longitudinal mscrodispersion in nonstationary flow fields
becomes anomalous and a Fick's regime is not achieved. This behavior of anomalous
macrodispersion is attributed to the effect of nonstationary hydraulic head fields
caused by the variation of hydraulic conductivity.

A macrodispersion coefficient in the mean flow direction can be defined by half

of the time derivative of Eq. (35b) as follows:
$$D_{11_y}(t) = \sigma_y^2 \lambda_y V \left[ 1 + \frac{9}{\Gamma^3} - \frac{3}{2}\frac{1}{\Gamma} + \frac{3}{8}\Gamma - e^{-\Gamma} \left( 1 + \frac{9}{\Gamma^3} + \frac{9}{\Gamma^2} + \frac{3}{\Gamma} \right) \right].$$
(41a)

This implies that the longitudinal macrodispersion coefficient at large time in
nonstationary flow fields can be approximated as
$$D_{11_y}(t) \approx \sigma_y^2 \lambda_y V \left( 1 + \frac{3}{8}\Gamma \right).$$
(41b)

That is, the longitudinal macrodispersion increases linearly with travel time at large



distances. Note that, in stationary flow fields, the longitudinal macrodispersion
coefficient approaches an asymptotic limit $D_{11_y} = \sigma_y^2 \lambda_y V$ at large time. Clearly,
applying the asymptotic macrodispersion coefficient (Eq. (41b)), which is appropriate
for macrodispersion in stationary flow fields, to the prediction of macrodispersion in
the downstream region at a large distance from the contamination source leads to a
significant underestimation of macrodispersion in nonstationary flow fields.

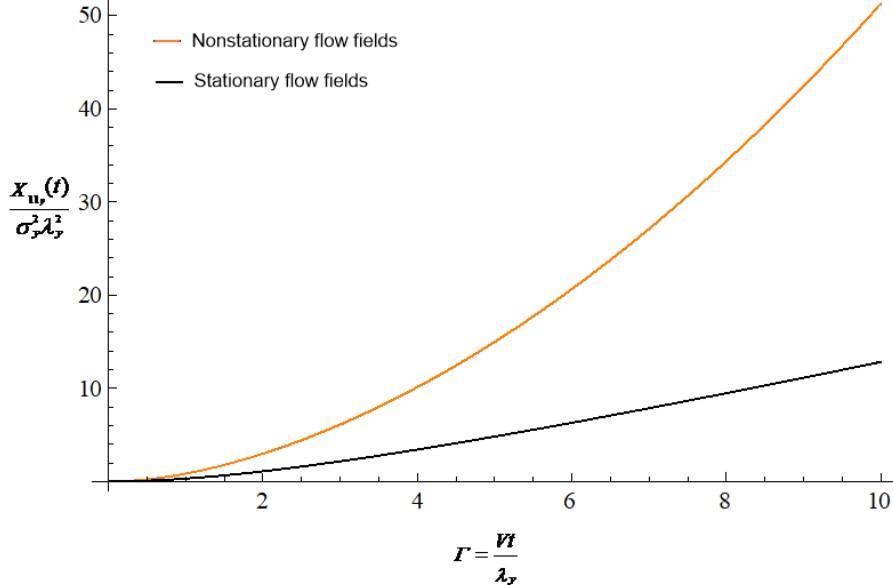


**Figure 3.** Comparison of the prediction of the solute longitudinal displacement
variance in Eq. (35b) in nonstationary flow fields with the prediction in Eq. (40b) in
stationary flow fields.

The behavior of the longitudinal displacement variance of solutes, affected by the

effect of variation of aquifer thickness field, in the nonstationary flow field (Eq. (35c))
and in the stationary flow field (Eq. (40c)) as a function of travel time is also presented
graphically in Fig. 4. This again demonstrates that the displacement variance grows
faster than linear with travel time and the longitudinal macrodispersion becomes
anomalous at large travel times. The corresponding longitudinal macrodispersion
coefficient is
$$D_{11_\beta}(t) = \sigma_\beta^2 \lambda_\beta V \left[ 1 + \frac{36}{\vartheta^3} - \frac{2}{\vartheta} + \frac{3}{2}\vartheta - e^{-\vartheta}\left(5 + \frac{36}{\vartheta^3} + \frac{36}{\vartheta^2} + \frac{16}{\vartheta} + 2\vartheta\right) \right],$$
(42a)

with the approximation at large times as
$$D_{11_\beta}(t) \approx \sigma_\beta^2 \lambda_\beta V \left(1 + \frac{3}{2}\vartheta\right).$$
(42b)

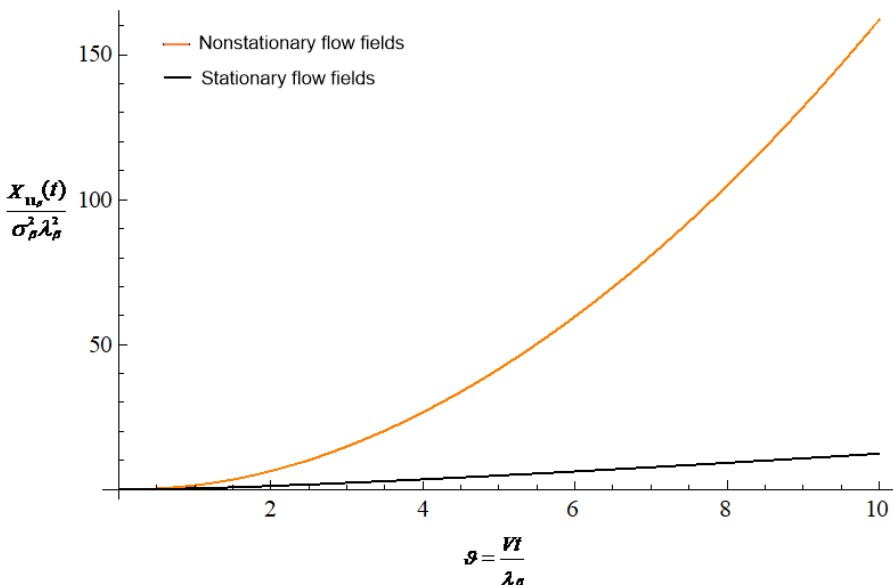


**Figure 4.** Comparison of the prediction of the solute longitudinal displacement
variance in Eq. (35c) in nonstationary flow fields with the prediction in Eq. (40c) in
stationary flow fields.



## 5    Conclusions


In this work, a theoretical stochastic methodology is developed to quantify the

displacement variance of an inert solute particle in heterogeneous confined aquifers

with variable thickness. This methodology relates solute displacement to the

Fokker-Planck equation through the two-dimensional depth-averaged solute mass

conservation equation. In contrast to previous stochastic studies of two-dimensional

solute transport problems, the variability of solute movement is caused not only by the

variability of log conductivity, but also by the variability of log thickness of confined

aquifer.

The two-dimensional stochastic groundwater flow equation for the

depth-averaged hydraulic head perturbation always has a 1-IRF solution when the log

hydraulic conductivity and log aquifer thickness fields are second-order stationary.

This leads to an unbounded increasing head semivariogram where no head covariance

exists. The nonstationarity of the hydraulic head leads to nonstationary flow velocity

fields and thus a nonlinear increase in longitudinal solute displacement with travel

time. That is, a Fick's regime is not achieved, and the longitudinal macrodispersion

becomes anomalous and increases linearly with travel time at large distances. It is also

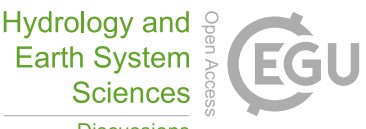

shown that the variability of solute displacement in the mean flow direction increases
with the variability of hydraulic conductivity and aquifer thickness.

**Appendix A: Expressions for the functions in Eqs. (26) and (28)**

$\Theta_1(a,b) = \dfrac{a}{r}\left[1 - e^{-r}(1+r)\right],$ (A1)
$\Theta_2(a,b) = -2\dfrac{a^2}{r^4} + \dfrac{1}{r^2} + e^{-r}\left[a^2(\dfrac{2}{r^4} + \dfrac{2}{r^3} + \dfrac{1}{r^2}) - \dfrac{1}{r^2} - \dfrac{1}{r}\right],$ (A2)
$\Theta_3(a,b) = ab\left[\dfrac{2}{r^4} - e^{-r}(\dfrac{2}{r^4} + \dfrac{2}{r^3} + \dfrac{1}{r^2})\right],$ (A3)
where $r^2 = a^2 + b^2$.

**Appendix B: Expressions for the functions in Eq. (29)**

$\Psi_1(a,b) = \dfrac{a^2 - b^2}{r^2}\left[\dfrac{1}{2} + \dfrac{e^{-r}(r^2 + 3r + 3) - 3}{r^2}\right] - Ei(r) + \ln(r) + e^{-r} - 1 + \gamma,$ (B1)
$\Psi_2(a,b) = \dfrac{1}{r^6}(a^4 + 6a^2 + 4a^2 b^2 + 3b^4 - 18b^2) + e^{-r}\left[-2\dfrac{a^6}{r^7} - a^4(\dfrac{6}{r^7} + \dfrac{4}{r^6}) - 6\dfrac{a^2}{r^6} - 4\dfrac{a^4 b^2}{r^7}\right.$
$\left. + 2a^2 b^2(\dfrac{6}{r^7} + \dfrac{1}{r^6}) - 2\dfrac{a^2 b^4}{r^7} + 6b^4(\dfrac{3}{r^7} + \dfrac{1}{r^6}) + 18\dfrac{b^2}{r^6}\right],$ (B2)
$\Psi_3(a,b) = \dfrac{1}{r^6}(a^4 - 18a^2 - b^4 + 6b^2) + e^{-r}\left[2\dfrac{a^6}{r^7} + 2a^4(\dfrac{9}{r^7} + \dfrac{4}{r^6}) + 18\dfrac{a^2}{r^6} + 4\dfrac{a^4 b^2}{r^7}\right.$
$\left. + 2\dfrac{a^2 b^4}{r^7} + 6a^2 b^2(\dfrac{2}{r^7} + \dfrac{1}{r^6}) - 2b^4(\dfrac{3}{r^7} + \dfrac{1}{r^6}) - 6\dfrac{b^2}{r^6}\right],$ (B3)
where $r^2 = a^2 + b^2$, $Ei$ is the exponential integral, and $\gamma$ is the Euler constant.





**Appendix C: Expressions for the functions in Eq. (30)**

$$\Xi_1(a,b) = -2\frac{9}{r^4} + \frac{1}{r^2} + e^{-r}[a^2(\frac{2}{r^4} + \frac{2}{r^3} + \frac{1}{r^2}) - \frac{1}{r^2} - \frac{1}{r}],$$  (C1)
$$\Xi_2(a,b) = -\frac{1}{2}\frac{1}{r^8}\Omega_1 + \frac{e^{-r}}{r^9}\Omega_2 + \frac{e^{-r}}{r^8}\Omega_3,$$  (C2)
where $r = (a^2 + b^2)^{1/2}$,
$$\Omega_1(a,b) = a^6 + 3a^2b^2(-36 + b^2) - 3b^4(-6 + b^2) + a^4(18 + 7b^2),$$  (C3)
$$\Omega_2(a,b) = 2a^8 + 9b^6 + a^6(9 - 2b^2) - 5a^4b^2(9 + 2b^2) - 3a^2b^4(15 + 2b^2),$$  (C4)
$$\Omega_3(a,b) = a^8 + 3b^4(3 + b^2) + a^6(5 + 2b^2) - 3a^2b^2(18 + 7b^2) + b^4(9 - 19b^2 + b^4),$$  (C5)

**Appendix D: Expressions for the functions in Eq. (31)**

$$\Delta_1(a,b) = 3a^8 + 4a^6(-1 + 3b^2) + b^4(72 - 4b^2 + 3b^4) + 4a^2b^4(-108 + 5b^2 + 3b^4) + 2a^4(36 + 10b^2 + 9b^4),$$  (D1)
$$\Delta_2(a,b) = -8a^8 + 4a^6[-18 - 8r + (8 + 2r)3b^2] + b^4[-72r - 4(8 + 8r)b^2 - 8b^4]$$
$$+ 4a^2b^2[108r + 5(18 + 8r)b^2 + (8 + 2r)b^4] + 2a^4[-36r + 10(18 + 8r)b^2 + (40 + 8r)b^4],$$  (D2)
$$\Delta_3(a,b) = a^8 + 4a^6b^2 + b^4(144 - 16b^2 + b^4) + 4a^2b^4(-216 + 8b^2 + b^4) + 6a^4(24 + 8b^2 + b^4),$$  (D3)
$$\Delta_4(a,b) = -8(3 + r)a^8 + 4a^6[-18(2 + r) + (12 - 2r)b^2] + b^4[-144r - 8(18 + 7r)b^2 - 8b^4]$$
$$+ 4a^2b^2[216r + 2(90 + 41r)b^2 + (20 + 2r)b^4] + 2a^4[-72r + 12(30 + 13r)b^2 + (80 + 4r)b^4],$$  (D4)
where $r = (a^2 + b^2)^{1/2}$.



*Data availability*. No data was used for the research described in the article.

*Author contributions*. C-MC: Conceptualization, Methodology, Formal analysis,
Writing - original draft preparation, Writing - review & editing.
C-FN: Conceptualization, Methodology, Formal analysis, Writing - original draft
preparation, Writing - review & editing, Supervision, Funding acquisition.
C-PL: Conceptualization, Methodology, Formal analysis, Writing - original draft
preparation, Writing - review & editing.
I-HL: Conceptualization, Methodology, Formal analysis, Writing - original draft
preparation, Writing - review & editing.

*Competing interests*. The authors declare that they have no conflict of interest.

*Acknowledgements*. Research leading to this paper has been partially supported by the
grant from the Taiwan Ministry of Science and Technology under the grants MOST
108-2638-E-008-001-MY2, MOST 110-2123-M-008-001-, MOST
110-2621-M-008-003-, and MOST 110-2811-M-008-533.

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
