# Peer review of "Technical note: Displacement variance of a solute particle in heterogeneous confined aquifers with random aquifer thickness fields"

_Hydrology and Earth System Sciences, 2022_

## Author Comment (AC1)

**Response to comments of Anonymous Referee 1**

We would like to thank the referee for the valuable comments and suggestions, which improved the quality of the paper. Below is our response to the comments and suggestions.

**Comment on hess-2022-298**

I have tried to read this paper multiple times now and every time find myself frustrated. I am highly litterate in terms of mathematically dense papers, but I found this paper next to impossible to make my way through. I do not usually write grumpy reviews, but this will be one. I have three major concerns that lead me to recommend that this paper be rejected.

(1) My first and likely biggest issue is going from equation (1) to (2). Any time you average and ADE equation like the one the authors have you will have a mean and fluctuation of the things that vary. In this case concentration, velocity and depending on the nature of the dispersion coefficient that also. Where are all of these gone? They don't just dissappear as it seems that they do in equation (2) - i.e. it's fine to say that the average of the fluctuation of concentration is zero, but the average of the product of concentration and velocity fluctuations is not. Indeed this is exactly what leads to things like macrodispersion and deviations from standard behaviors. Where have these gone here? There is no discussion of them and none of the assumptions I see in the problem setup suggest they do not exist or are negligible. This is the starting point of the paper and frankly makes me feel like the authors are departing from a faulty point from the getgo.

**Response**

a. The derivation of Eq. (1) to Eq. (2) was presented in Holly (1975). Equation (2) has been widely used to analyze problems related to solute transport by fluid flow (e.g., Zerihun et al. 2005, Baek et al. 2006, Chavez et al. 2014).

   Zerihun, D., Furman, A., Warrick, A. W., and Sanchez, C. A.: Coupled surface-subsurface solute transport model for irrigation borders and basins. I. Model development, J. Irrig. Drain. Eng., ASCE 131(5), 396-406, 2005.
   Baek, K. O., Seo, I. W., and Jeong, S. J.: Evaluation of dispersion coefficients in meandering channels from transient tracer tests. J. Hydraul. Eng., ASCE 132 (10), 1021-1032, 2006.
   Chavez, C., Fuentes, C., Brambila, F., and Castañeda, A.: Numerical solution of the advection-dispersion equation: Application to the agricultural drainage, J. Agric. Sci. Technol., 16(6), 1357-1388, 2014.

b. We apologize for not mentioning Holly's (1975) idea in developing Eq. (2) regarding the average of the product of concentration and velocity fluctuations. Holly (1975) considered the mixing of the contaminant plume over depth in natural channels to be complete, so that the fluctuations around the depth-averaged concentration are relatively small. Then the average of the product of concentration and velocity fluctuations can be considered to be absorbed into the gradient transport terms in Eq. (2).
   A note is added to the manuscript mentioning this as follows:
   "In developing Eq. (2), it is assumed that the contaminant plume in confined aquifers is well mixed over depth, so that variations around the depth-averaged concentration are relatively small. Then the average of the product of concentration and velocity fluctuations can be assumed to be absorbed in the gradient transport terms in Eq. (2)"

c. It can be clearly seen (or verified) that Eq. (2) for flow in aquifers of uniform thickness (i.e., $B(x_1,x_2)$ = constant) reduces to the traditional two-dimensional advection-dispersion equation for solute transport in confined aquifers, with the flow fields characterized by the aquifer transmissivity fields instead of the hydraulic conductivity fields.

(2)  As I noted I am someone who writes and reads a lot of papers with pretty dense and complex mathematics in it, but I found a lot of what the authors present extremely hard to follow, where in some places there is abundant detail and in others serious gaps.

**Response**

Please clarify. We will do our best to change it.

(3)  Last but not least, even if everything is right (which I cannot verify) I struggle to see the real importance of this paper and thus am hesitant to see it published in such a high level journal such as HESS which is one of the top journals in our field. Much of the paper feels a little archaic in nature and while I love theoretical papers with full mathematics I also feel that something clear should be gained by ellaborating it and I just do not see that here.

**Response**

a. Natural confined aquifers at the regional scale often exhibit nonuniform thickness. In the traditional approach to regional groundwater flow problems, the effects of aquifer thickness variations are implicit in aquifer transmissivity term. Therefore, it is very difficult to use the traditional approach to assess the influence of thickness on the flow field and thus the displacement of solutes.

b. In this work, the relationship between the two-dimensional depth-averaged solute conservation equation and the Fokker-Planck equation is used to relate the effects of aquifer thickness variations to flow field variations. In this way, a definite relationship can be established between the thickness variation and the solute displacement variation. This work shows that variability in aquifer thickness can lead to nonstationarity in hydraulic head fields and thus to nonstationary flow velocity fields and anomalous longitudinal dispersion. The work also shows that neglecting the variability of aquifer thickness when predicting the longitudinal displacement of solutes at large times can lead to a significant underestimation of longitudinal dispersion. To our knowledge, the analysis of the influence of the variability of the thickness of the aquifer on the longitudinal displacement of the solute within the framework of stochastics has not yet been presented in the literature.

---

## Author Comment (AC2)

**Response to comments of Anonymous Referee 1**

We would like to thank the referee for the valuable comments and suggestions, which improved the quality of the paper. Below is our response to the comments and suggestions.

**Comment on hess-2022-298**

I have tried to read this paper multiple times now and every time find myself frustrated. I am highly litterate in terms of mathematically dense papers, but I found this paper next to impossible to make my way through. I do not usually write grumpy reviews, but this will be one. I have three major concerns that lead me to recommend that this paper be rejected.

(1) My first and likely biggest issue is going from equation (1) to (2). Any time you average and ADE equation like the one the authors have you will have a mean and fluctuation of the things that vary. In this case concentration, velocity and depending on the nature of the dispersion coefficient that also. Where are all of these gone? They don't just disappear as it seems that they do in equation (2) - i.e. it's fine to say that the average of the fluctuation of concentration is zero, but the average of the product of concentration and velocity fluctuations is not. Indeed this is exactly what leads to things like macrodispersion and deviations from standard behaviors. Where have these gone here? There is no discussion of them and none of the assumptions I see in the problem setup suggest they do not exist or are negligible. This is the starting point of the paper and frankly makes me feel like the authors are departing from a faulty point from the getgo.

**Response**

    a. The derivation of Eq. (1) to Eq. (2) was presented in Holly (1975). Equation (2) also appears in the textbook by Fischer et al. (1979). In addition, Eq. (2) has been widely used to analyze problems related to solute transport by fluid flow (e.g., Zerihun et al. 2005, Baek et al. 2006, Chavez et al. 2014).

        Zerihun, D., Furman, A., Warrick, A. W., and Sanchez, C. A.: Coupled surface-subsurface solute transport model for irrigation borders and basins. I. Model development, J. Irrig. Drain. Eng., ASCE 131(5), 396-406, 2005.

        Baek, K. O., Seo, I. W., and Jeong, S. J.: Evaluation of dispersion coefficients in meandering channels from transient tracer tests. J. Hydraul. Eng., ASCE 132

(10), 1021-1032, 2006.

Chavez, C., Fuentes, C., Brambila, F., and Castañeda, A.: Numerical solution of the advection-dispersion equation: Application to the agricultural drainage, J. Agric. Sci. Technol., 16(6), 1357-1388, 2014.

b. We apologize for not mentioning Holly's (1975) idea in developing Eq. (2) regarding the average of the product of concentration and velocity fluctuations. Holly (1975) considered the mixing of the contaminant plume over depth in natural channels to be complete, so that the fluctuations around the depth-averaged concentration are relatively small. Then the average of the product of concentration and velocity fluctuations can be considered to be absorbed into the gradient transport terms in Eq. (2).

A note is added to the manuscript mentioning this as follows (Line 446 on page 26):

"In developing Eq. (A2), it is assumed that the contaminant plume in confined aquifers is well mixed over depth, so that variations around the depth-averaged concentration are relatively small (Holly, 1975). Then the average of the product of concentration and velocity fluctuations can be assumed to be absorbed in the gradient transport terms in Eq. (A2)"

c. It can be clearly seen (or verified) that Eq. (2) for flow in aquifers of uniform thickness (i.e., $B(x_1, x_2)$ = constant) reduces to the traditional two-dimensional advection-dispersion equation for solute transport in confined aquifers, with the flow fields characterized by the aquifer transmissivity fields instead of the hydraulic conductivity fields.

(2)   As I noted I am someone who writes and reads a lot of papers with pretty dense and complex mathematics in it, but I found a lot of what the authors present extremely hard to follow, where in some places there is abundant detail and in others serious gaps.

**Response**

The structure of the manuscript was fundamentally changed to make it clear and readable.

a. A brief preview of this work is added on page 5 (Line 74) as

"In the present work, the convection velocity of solute particles is first developed based on the relationship between the two-dimensional depth-averaged solute mass conservation equation and the Fokker-Planck equation, so that the convection velocity can explicitly reflect the effects

of hydraulic conductivity and aquifer thickness. Using the perturbation approach to solute convection velocity, the covariance function of solute convection velocity is then developed, which allows a general expression for the variance of the displacement of a solute particle in the mean flow direction to be developed. A closed-form expression for the solute displacement variance is also developed for the case where solute transport is dominated by advection and the random fields of log conductivity and log thickness of the confined aquifer are second-order stationary. Finally, the influence of variations in log hydraulic conductivity and log aquifer thickness on the variability of solution displacement is analyzed."

b. To facilitate understanding, we have restructured the manuscript so that the main text of the manuscript focuses on the step-by-step development of the variance of the solute displacement, while many details of the mathematical derivations related to the flow fields have been moved to Appendices A and B, such as the detailed solute convection velocity derivation and the cross-covariance and covariance functions of the flow velocity fields. **For details, please see the revised manuscript.**

c. Further insight is provided to better understand the idea behind Figures 1-2 (Line 280 on page 17):

"When taking samples from a field, one obtains a histogram from which a certain value of the variance can always be calculated. However, for many phenomena, the experimental variance is actually a function of the field. In particular, it increases as the field increases, i.e., many phenomena have an almost unlimited capacity of dispersion and cannot be adequately described by ascribing to them a finite a priori variance. In this case, the use of the semivariogram is an appropriate way to measure the variability of the variation."

(3)  Last but not least, even if everything is right (which I cannot verify) I struggle to see the real importance of this paper and thus am hesitant to see it published in such a high level journal such as HESS which is one of the top journals in our field. Much of the paper feels a little archaic in nature and while I love theoretical papers with full mathematics I also feel that something clear should be gained by ellaborating it and I just do not see that here.

**Response**

a. To make it clear that this is a new and original work and the results of this work are important, the novelty of this study is added in the **Introduction** section (Line 59 on page 5) as follows:

"The traditional approach to regional groundwater flow problems introduces the transmissivity parameter to describe the ability of a confined aquifer to transmit water throughout its saturated thickness. The effect of the thickness of the aquifer is implicitly reflected in the transmissivity parameter. It is very difficult to assess the effect of thickness on the flow field and thus on solute transport at a regional scale. The stochastic approach presented here provides an efficient and rational way to analyze flow and solute transport fields affected by the non-uniform thickness of confined aquifers, which has not been previously presented in the literature. This work shows that variability in aquifer thickness can lead to nonstationarity in hydraulic head fields and thus to nonstationary flow velocity fields and anomalous longitudinal dispersion. This implies that neglecting the variability of aquifer thickness when predicting the longitudinal displacement of solutes at large times can lead to a significant underestimation of longitudinal dispersion. The stochastic theory presented here improves quantification of the variance of the solute displacement in natural confined aquifers of random thickness fields."

b. To our knowledge, the analysis of the influence of the variability of the thickness of the aquifer on the longitudinal displacement of the solute within the framework of stochastics has not yet been presented in the literature.

We believe that the manuscript is of the quality required for publication and should be of interest to many readers in *Hydrology and Earth System Science*.

---

## Author Comment (AC3)

**Response to comments of Anonymous Referee 2**

We would like to thank the referee for the valuable comments and suggestions, which improved the quality of the paper. Below is our response to the comments and suggestions.

**Comment on hess-2022-298**

My impression is that the novelty that this work brings is scarce. Probably this is partially due to the presentation of the derivation and the results that is quite confusing. I suggest the authors revise completely the paper by improving the description of the mathematical approach and by relating it to the state-of-art to underline the advancements introduced by the study. I also suggest revising the figures that presently are of poor quality and not so explicative. I ask the authors to rethink the graphical representation of the results and to add more graphical insights to help the comprehension. In summary, I suggest major revisions to the manuscript even if I'm aware that the sum of all the revisions would lead to a very different version of the manuscript.

**Response**

1. To make it clear that this is a new and original work and the results of this work are important, the novelty of this study is added in the introduction section as follows (Line 59 on page 5):

   "The traditional approach to regional groundwater flow problems introduces the transmissivity parameter to describe the ability of a confined aquifer to transmit water throughout its saturated thickness. The effect of the thickness of the aquifer is implicitly reflected in the transmissivity parameter. It is very difficult to assess the effect of thickness on the flow field and thus on solute transport at a regional scale. The stochastic approach presented here provides an efficient and rational way to analyze flow and solute transport fields affected by the non-uniform thickness of confined aquifers, which has not been previously presented in the literature. This work shows that variability in aquifer thickness can lead to nonstationarity in hydraulic head fields and thus to nonstationary flow velocity fields and anomalous longitudinal dispersion. This implies that neglecting the variability of aquifer thickness when predicting the longitudinal displacement of solutes at large times can lead to a significant underestimation of longitudinal dispersion. The stochastic theory presented here improves quantification of the variance of the solute displacement in natural confined aquifers of random thickness fields."

2. In order to make the manuscript clear and easy to read, we made major adjustments to the structure of the manuscript.

    a. A brief preview of this work is added on page 5 (Line 74) as

        "In the present work, the convection velocity of solute particles is first developed based on the relationship between the two-dimensional depth-averaged solute mass conservation equation and the Fokker-Planck equation, so that the convection velocity can explicitly reflect the effects of hydraulic conductivity and aquifer thickness. Using the perturbation approach to solute convection velocity, the covariance function of solute convection velocity is then developed, which allows a general expression for the variance of the displacement of a solute particle in the mean flow direction to be developed. A closed-form expression for the solute displacement variance is also developed for the case where solute transport is dominated by advection and the random fields of log conductivity and log thickness of the confined aquifer are second-order stationary. Finally, the influence of variations in log hydraulic conductivity and log aquifer thickness on the variability of solution displacement is analyzed."

    b. To facilitate understanding, we have restructured the manuscript so that the main text of the manuscript focuses on the step-by-step development of the variance of the solute displacement, while many details of the mathematical derivations related to the flow fields have been moved to Appendices A and B, such as the detailed solute convection velocity derivation and the cross-covariance and covariance functions of the flow velocity fields. **For details, please see the revised manuscript.**

3. Further insight is provided to better understand the idea behind Figures 1-2 (Line 280 on page 17):

        "When taking samples from a field, one obtains a histogram from which a certain value of the variance can always be calculated. However, for many phenomena, the experimental variance is actually a function of the field. In particular, it increases as the field increases, i.e., many phenomena have an almost unlimited capacity of dispersion and cannot be adequately described by ascribing to them a finite a priori variance. In this case, the use of the semivariogram is an appropriate way to measure the variability of the variation."

Yes, major revisions to the manuscript result in a very different version of the manuscript.

---

## Author Comment (AC5)

[revised manuscript text omitted]

The traditional approach to regional groundwater flow problems introduces the transmissivity parameter to describe the ability of a confined aquifer to transmit water throughout its saturated thickness. The effect of the thickness of the aquifer is implicitly reflected in the transmissivity parameter. It is very difficult to assess the effect of thickness on the flow field and thus on solute transport at a regional scale.

The stochastic approach presented here provides an efficient and rational way to analyze flow and solute transport fields affected by the non-uniform thickness of confined aquifers, which has not been previously presented in the literature. This work shows that variability in aquifer thickness can lead to nonstationarity in hydraulic head fields and thus to nonstationary flow velocity fields and anomalous longitudinal dispersion. This implies that neglecting the variability of aquifer thickness when predicting the longitudinal displacement of solutes at large times can lead to a significant underestimation of longitudinal dispersion. The stochastic theory presented here improves quantification of the variance of the solute displacement in natural confined aquifers of random thickness fields.

In the present work, the convection velocity of solute particles is first developed based on the relationship between the two-dimensional depth-averaged solute mass conservation equation and the Fokker-Planck equation, so that the convection velocity can explicitly reflect the effects of hydraulic conductivity and aquifer thickness. Using the perturbation approach to solute convection velocity, the covariance function of solute convection velocity is then developed, which allows a general expression for the variance of the displacement of a solute particle in the mean flow direction to be developed. A closed-form expression for the solute displacement variance is also developed for the case where solute transport is dominated by advection and the random fields of log conductivity and log thickness of the confined aquifer are second-order stationary. Finally, the influence of variations in log hydraulic conductivity and log aquifer thickness on the variability of solution displacement is analyzed.

**2     Mathematical formulation of the problem**

Consider here the steady flow of a fluid carrying an inert solute through a heterogeneous confined aquifer with variable thickness. When constituents are well mixed throughout the thickness of the aquifer (depth of flow) and fluid flow through an aquifer occurs on a regional scale, with the lateral extent of the formation much greater than the thickness of the formation, it is appropriate to view the flow and solute transport processes as essentially two-dimensional. In this work, the two-dimensional solute transport process in heterogeneous confined aquifers is quantified by using moments of solute particle displacement in the Lagrangian framework (e.g., Dagan,

1982; 1984), where the particle displacement can be defined as

$$\frac{dX}{dt} = V_c$$    (1)

In Eq. (1), $X$ (= $(X_1, X_2)$) is the displacement and $V_c$ (= $(V_{c_1}, V_{c_2})$) is the convection velocity of the solute particle.

The displacement of the solute particles in Eq. (1) consists of two components:

one originates from convection through the fluid and the other is associated with the transport process at the pore scale. This means that the statistical moments of particle displacement cannot be determined directly from the statistical moments of flow velocity. The convection velocity of the solute particle in Eq. (1) can be obtained from the relationship between the two-dimensional depth-averaged equation for the conservation of solute mass and the Fokker-Planck equation as follows:

$$\frac{dX_i}{dt} = \frac{1}{n}\tilde{q}_i(X) + [\frac{1}{n}\tilde{D}_i(X)\frac{\partial}{\partial x_i}\ln B(X) + \frac{1}{n}\frac{\partial}{\partial x_i}\tilde{D}_i(X)] + \sqrt{\frac{2}{n}\tilde{D}_i(X)}\frac{dW}{dt} \qquad i = 1,2.$$    (2)

where $n$ is the porosity, $\tilde{D}_i$, and $\tilde{q}_i$ represent the depth-averaged dispersion coefficient and depth-averaged specific discharge in the $x_i$ direction, respectively, $B$ is the thickness of a confined aquifer, and $W$ denotes a Wiener process. The details of the development of Eq. (2) are given in Appendix A.

From the right-hand side of Eq. (2), it can be seen that the first term represents the convection velocity of the flow, the second and third terms are associated with pore-scale dispersion, which includes the effects of local heterogeneity of aquifer thickness and dispersion coefficient, respectively, and the last term is associated with a

Brownian motion type diffusion process. Equation (2) provides a basic basis for evaluating the statistical moments of solute particle displacement.

In this study, the fields (or processes) of hydraulic conductivity $K(x_1,x_2)$ and thickness of the confined aquifer $B(x_1,x_2)$ are considered spatially random, and therefore a random flow field and a random particle displacement field. It is also assumed that the mean fluid flow is uniform and unidirectional in the $x_1$-direction (i.e.,

$\langle X \rangle = (\langle X_1 \rangle, 0)$) and that the spatial variation of the depth-averaged dispersion coefficients and the Brownian motion type diffusion process are negligible. This simplifies Eq. (2) to

$$\frac{dX_i}{dt} = \frac{1}{n}\tilde{q}_i(X) + \frac{1}{n}\tilde{D}_i\frac{\partial}{\partial x_i}\ln B(X) \qquad i = 1,2. \tag{3}$$

Note that the assumption of uniform mean flow in the $x_1$-direction implies that the gradient of the mean depth-averaged hydraulic head is constant in the $x_1$-direction and zero in the $x_2$-direction (Chang et al. 2021).

By analogy with Butera and Tanda (1999), extending Eq. (3) in Taylor series around $\langle X \rangle$ in the $x_1$-direction yields

$$\frac{dX_1}{dt} = \frac{1}{n}\tilde{D}_1\left[\frac{\partial \Phi(\langle X_1 \rangle, 0)}{dx_1} + \frac{d^2\Phi(\langle X_1 \rangle, 0)}{dx_1^2}X_1' + \frac{d\beta(\langle X_1 \rangle, 0)}{dx_1}\right] + \langle \tilde{v}_1 \rangle + v_1(\langle X_1 \rangle, 0), \tag{4}$$

where $X_1' = X_1 - <X_1>$, $\Phi = <\ln B>$, $\beta = \ln B - <\ln B>$, $v_1 = \tilde{v}_1 - <\tilde{v}_1>$, $<\tilde{v}_1>$ = constant, and $\tilde{v}_1 = \tilde{q}_1/n$. Note that due to the assumption of uniform mean flow in the

$x_1$-direction, the term $\dfrac{d<\tilde{v}_1>}{dx_1} X_1'$ has been removed from Eq. (4). Equation (4) reveals that

$\dfrac{d<X_1>}{dt} = \dfrac{1}{n}\tilde{D}_1 \dfrac{d\Phi(<X_1>,0)}{dx_1} + <\tilde{v}_1>,$ (5a)

$\dfrac{dX_1'}{dt} - \dfrac{\tilde{D}_1}{n}\dfrac{d^2\Phi(<X_1>,0)}{dx_1^2}X_1' = \dfrac{\tilde{D}_1}{n}\dfrac{d\beta(<X_1>,0)}{dx_1} + v_1(<X_1>,0).$ (5b)

Equations (5a) and (5b) describe the mean and fluctuation, respectively, of the displacement of the solute particles. By the solution of Eq. (5), the variance of the solute displacement in the $x_1$-direction (the mean flow direction) can be evaluated in the frame, (e.g., Dagan, 1984; 1989)

$X_{11}(t) = <X_1'(t)X_1'(t)>.$ (6)

It is important to recognize the validity of the assumption of a first order perturbation of $X_1$. The first-order approximation for representing the depth-averaged hydraulic head perturbation, and hence the solute displacement perturbation, should be applied to porous formations where the standard deviation of the random fluctuations of the log hydraulic conductivity is less than 1. However, Zhang and

Winter (1999) report in a Monte Carlo simulation study that it is accurate for the solutions of the head moment for the value of the variance of the log conductivity of up to 4.38. A similar finding from comparing moments of hydraulic head with results of numerical Monte Carlo simulations is also reported in Guadagnini and Neuman (1999) for highly heterogeneous media with a variance of log conductivity from 2 to

4.

In the case where the thickness of the aquifer is a slowly spatially varying process (e.g., a second-order stationary process), the terms $d\Phi/dx_1$ and $d^2\Phi/dx_1^2$ in Eq. (5) may be neglected, and, consequently, Eq. (5) reduces to

$$\frac{d<X_1>}{dt}=<\tilde{v}_1>, \tag{7a}$$

$$\frac{dX_1^{'}}{dt}=\frac{\tilde{D}_1}{n}\frac{d\beta(<X_1>,0)}{dx_1}+v_1(<X_1>,0). \tag{7b}$$

Equation (7b) implies that the variability of the particle displacement is determined by the gradient of the variation of the aquifer thickness fields and the variability of the flow velocity. Note that when flowing through a confined aquifer with variable thickness, the variability in flow velocity is influenced by both the variation in log conductivity and log thickness fields (Chang et al., 2021). This means that the variability of $v_1$ in Eq. (7b) depends on both the variation of log conductivity and log aquifer thickness.

Using the solution of Eq. (7),

$$X_1^{'}(t)=\int_0^t[\frac{\tilde{D}_1}{n}\frac{d\beta}{dx_1}(<\tilde{v}_1>s,0)+v_1(<\tilde{v}_1>s,0)]ds, \tag{8}$$

the variance of the solute displacement in the mean flow direction in Eq. (6) results in

$$X_{11}(t) = \int_0^t \int_0^t \left[ \frac{\tilde{D}_1^2}{n^2} < \frac{\partial \beta(\xi)}{\partial \xi_1} \Big|_{\ell_1} \frac{\partial \beta(\zeta)}{\partial \zeta_1} \Big|_{\ell_2} + \frac{\tilde{D}_1}{n} < \frac{\partial \beta(\xi)}{\partial \xi_1} \Big|_{\ell_1} v_1(\zeta) \Big|_{\ell_2} > \right.$$

$$\left. + \frac{\tilde{D}_1}{n} < v_1(\xi) \Big|_{\ell_1} \frac{\partial \beta(\zeta)}{\partial \zeta_1} \Big|_{\ell_2} > + < v_1(\xi) \Big|_{\ell_1} v_1(\zeta) \Big|_{\ell_2} > \right] dS_1 dS_2, \qquad (9)$$

where $\xi = (\xi_1, \xi_2)$, $\zeta = (\zeta_1, \zeta_2)$, $\ell_1 = (<\tilde{v}_1 > S_1, 0)$, and $\ell_2 = (<\tilde{v}_1 > S_2, 0)$. To arrive at Eq.

(9), the solute particle was assumed to begin its motion at location $x_1 = 0$ and time $t =$

0.

To proceed with the evaluation of solute displacement in the $x_1$ direction, the following section develops the statistics of the flow fields in Eq. (9) for the case where both the variations in hydraulic conductivity and the thickness of the confined aquifer are considered to be second-order stationary processes and the random processes of hydraulic conductivity and aquifer thickness are statistically independent.

**3   Statistics of the flow fields**

Chang et al. (2021) develop the differential equations for the flow fields (Eqs. (6) and (12) of Chang et al., 2021) in a confined aquifer with variable thickness based on a hydraulic approach to flow in aquifers (Bear, 1979; Bear and Cheng, 2010). On this basis, under the condition of steady-state flow, the equation for the depth-averaged specific discharge about the mean, keeping only first-order terms in the perturbations, take the following form

$\quad q_i = \bar{q}[(y+\beta)\delta_{1i} - \dfrac{1}{J}\dfrac{\partial h}{\partial x_i}] \qquad i = 1,2,$ (10a)

where $\quad h = \tilde{h} - <\tilde{h}>$, $\tilde{h}$ is the depth-averaged hydraulic head, $J = -d<\tilde{h}>/dx_1 \,(=$

constant), $y = \ln K - Y$, $K$ is the hydraulic conductivity, $Y = <\ln K>$, $q_i = \tilde{q}_i - <\tilde{q}_i>$,

$\bar{q} = <\tilde{q}_1> = e^Y J$, and the equation describing the depth-averaged head perturbation is of the form

$\quad \dfrac{\partial^2 h}{\partial x_i^2} = J[\dfrac{\partial y}{\partial x_1} + 2\dfrac{\partial \beta}{\partial x_1}] \qquad i = 1,2.$ (10b)

Equation (10) shows that the variations in log-hydraulic conductivity and log-aquifer thickness appear as forcing terms that produce the variations in depth-averaged head and hence the variations in depth-averaged specific discharge.

It follows from Eq. (10) that the terms for the statistics of the flow fields in Eq. (9), such as the covariance function for the log-aquifer thickness gradient, the cross-correlation between the log-aquifer thickness gradient and the depth-averaged flow velocity, and the covariance function for the depth-averaged flow velocity process, can be evaluated using the spectral representation theorem as follows:

$\quad <\dfrac{\partial \beta(\xi)}{\partial \xi_1}\dfrac{\partial \beta(\zeta)}{\partial \zeta_1}> = \dfrac{\partial^2}{\partial \xi_1 \partial \zeta_1}C_{\beta\beta}(\xi,\zeta),$ (11)

$\quad <\dfrac{\partial \beta(\xi)}{\partial \xi_1}v_1(\zeta)> = VJ\dfrac{\partial}{\partial \xi_1}C_{\beta\beta}(\zeta,\xi) - V\dfrac{\partial^2}{\partial \xi_1 \partial \zeta_1}C_{\beta h_\beta}(\xi,\zeta),$ (12a)

$\quad <v_1(\xi)\dfrac{\partial \beta(\zeta)}{\partial \zeta_1}> = VJ\dfrac{\partial}{\partial \zeta_1}C_{\beta\beta}(\xi,\zeta) - V\dfrac{\partial^2}{\partial \xi_1 \partial \zeta_1}C_{\beta h_\beta}(\zeta,\xi),$ (12b)

$\quad \dfrac{<v_i(\xi)v_j(\zeta)>}{V^2} = [C_{yy}(\xi,\zeta) + C_{\beta\beta}(\xi,\zeta)]\delta_{1i}\delta_{1j} - \dfrac{1}{J}\dfrac{\partial}{\partial \zeta_j}[C_{yh_y}(\xi,\zeta) + C_{\beta h_\beta}(\xi,\zeta)]\delta_{1i}$

$\qquad\qquad -\dfrac{1}{J}\dfrac{\partial}{\partial \xi_i}[C_{yh_y}(\zeta,\xi) + C_{\beta h_\beta}(\zeta,\xi)]\delta_{1j} - \dfrac{1}{J^2}\dfrac{\partial \gamma_h(\xi,\zeta)}{\partial \xi_i \partial \zeta_j},$ (13)

209 where $V = \overline{q}/n = e^Y J/n$, $C_{yy}$ and $C_{\beta\beta}$ are the $\ln K$ and $\ln B$ covariance functions,

210 respectively, $C_{yh_y}$ is the covariance of $\ln K$ process with the head process, $C_{\beta h_\beta}$ is the

211 covariance of $\ln B$ process with the head process, and $\gamma_h$ is the semivariogram of the

212 head process, defined as

213 $$\gamma_h(\xi,\zeta) = \gamma_{h_y}(\xi,\zeta) + \gamma_{h_\beta}(\xi,\zeta) = \frac{1}{2}\left\{ <[h_y(\xi) - h_y(\zeta)]^2> + <[h_\beta(\xi) - h_\beta(\zeta)]^2> \right\}. \quad (14)$$

214 Note that $C_{yh_y}$, $C_{\beta h_\beta}$, and $\gamma_h$ in Eqs. (12) and (13) can be calculated using the

215 representation theorem for the depth-averaged head perturbation $h$ (the perturbation

216 solution of equation (10b)).

**218 4 Results and discussion**

220 To simplify the analysis of the variation of log-aquifer thickness on the variability of

221 the solute displacement, this study considers the case where the local dispersivity is

222 very small compared to the integral scales for the $\ln K$ and $\ln B$ processes, so that the

223 solute dispersion is mainly caused by the spatial variability of hydraulic conductivity

224 and thickness of confined aquifer. That is, solute dispersion occurs in situations where

225 advection dominates and solute particles do not transfer across streamlines. Therefore,

226 Eq. (9) can be simplified to

$$X_{11}(t)=\int_0^t\int_0^t <v_1(\xi)\big|_{\ell_1}\, v_1(\zeta)\big|_{\ell_2}> dS_1 dS_2 .$$ (15)

That is, the variance of the solute displacement in the mean flow direction can only be determined with Eqs. (15) and (13). There are numerous studies in the literature on solute transport under advection-dominated conditions, e.g., Dagan (1984), Rubin and

Bellin (1994), Butera et al. (2009), Cvetkovic (2016), Ciriello and Barros (2020), etc.

To determine the covariance function of the depth-averaged flow velocity, and thus the variance of solute displacement, it is assumed that the hydraulic conductivity and the thickness of the aquifer fields are lognormally distributed and characterized by the isotropic exponential covariance, i.e. (e.g., Dagan, 1984; Gelhar, 1993; Bailey and

Baù, 2012)

$$C_{yy}(\xi,\zeta)=\sigma_y^2\exp[-\frac{|\xi-\zeta|}{\lambda_y}],$$ (16a)

$$C_{\beta\beta}(\xi,\zeta)=\sigma_\beta^2\exp[-\frac{|\xi-\zeta|}{\lambda_\beta}],$$ (16b)

where $\sigma_y^2$ and $\sigma_\beta^2$ are the variances of $y$ and $\beta$, respectively, $\lambda_y$ and $\lambda_\beta$ are the integral scales of $\ln K$ and $\ln B$ fields, respectively. The corresponding spectra, which result from the inverse Fourier transform of Eq. (16), are as follows:

$$S_{yy}(R_1,R_2)=\frac{\sigma_y^2}{2\pi}\frac{\lambda_y^2}{[1+\lambda_y^2(R_1^2+R_2^2)]^{3/2}},$$ (17a)

$$S_{\beta\beta}(R_1,R_2)=\frac{\sigma_\beta^2}{2\pi}\frac{\lambda_\beta^2}{[1+\lambda_\beta^2(R_1^2+R_2^2)]^{3/2}} .$$ (17b)

**4.1 Covariance of flow velocity in the $x_1$-direction**

Once the spectrum forms of the ln$K$ and ln$B$ fields are selected, the cross-correlation between the ln$K$ perturbation and the perturbation in the depth-averaged head, $C_{yh_y}$, the cross-correlation between the ln$B$ perturbation and the perturbation in the depth-averaged head, $C_{\beta h_\beta}$, and the semivariogram of the depth-averaged process, $\gamma_h$, can be determined as follows:

$$C_{yh_y}(\boldsymbol{\xi},\boldsymbol{\zeta})=\sigma_y^2\lambda_y J\left[\Theta_1(\frac{\xi_1}{\lambda_y},\frac{\xi_2}{\lambda_y})-\frac{\zeta_1}{\lambda_y}\Theta_2(\frac{\xi_1}{\lambda_y},\frac{\xi_2}{\lambda_y})+\frac{\zeta_2}{\lambda_y}\Theta_3(\frac{\xi_1}{\lambda_y},\frac{\xi_2}{\lambda_y})-\Theta_1(\frac{\rho_1}{\lambda_y},\frac{\rho_2}{\lambda_y})\right],\tag{18}$$

$$C_{\beta h_\beta}(\boldsymbol{\xi},\boldsymbol{\zeta})=2\sigma_y^2\lambda_\beta J\left[\Theta_1(\frac{\xi_1}{\lambda_\beta},\frac{\xi_2}{\lambda_\beta})-\frac{\zeta_1}{\lambda_\beta}\Theta_2(\frac{\xi_1}{\lambda_\beta},\frac{\xi_2}{\lambda_\beta})+\frac{\zeta_2}{\lambda_\beta}\Theta_3(\frac{\xi_1}{\lambda_\beta},\frac{\xi_2}{\lambda_\beta})-\Theta_1(\frac{\rho_1}{\lambda_\beta},\frac{\rho_2}{\lambda_\beta})\right],\tag{19}$$

$$\gamma_h(\boldsymbol{\xi},\boldsymbol{\zeta})=\gamma_{h_y}(\boldsymbol{\xi},\boldsymbol{\zeta})+\gamma_{h_\beta}(\boldsymbol{\xi},\boldsymbol{\zeta}),\tag{20a}$$

$$\gamma_{h_y}(\boldsymbol{\xi},\boldsymbol{\zeta})=\frac{1}{2}\sigma_y^2\lambda_y^2 J^2\left\{\frac{3}{8}\frac{\rho_1^2}{\lambda_y^2}+\frac{1}{8}\frac{\rho_2^2}{\lambda_y^2}+\Psi_1(\frac{\rho_1}{\lambda_y},\frac{\rho_2}{\lambda_y})+\frac{\rho_1}{\lambda_y}\left[-\frac{\xi_1}{\lambda_y}\Psi_2(\frac{\xi_1}{\lambda_y},\frac{\xi_2}{\lambda_y})+\frac{\zeta_1}{\lambda_y}\Psi_2(\frac{\zeta_1}{\lambda_y},\frac{\zeta_2}{\lambda_y})\right]\right.$$

$$\left.+\frac{\rho_2}{\lambda_y}\left[\frac{\xi_2}{\lambda_y}\Psi_3(\frac{\xi_1}{\lambda_y},\frac{\xi_2}{\lambda_y})-\frac{\zeta_2}{\lambda_y}\Psi_3(\frac{\zeta_1}{\lambda_y},\frac{\zeta_2}{\lambda_y})\right]\right\},\tag{20b}$$

$$\gamma_{h_\beta}(\boldsymbol{\xi},\boldsymbol{\zeta})=2\sigma_\beta^2\lambda_\beta^2 J^2\left\{\frac{3}{8}\frac{\rho_1^2}{\lambda_\beta^2}+\frac{1}{8}\frac{\rho_2^2}{\lambda_\beta^2}+\Psi_1(\frac{\rho_1}{\lambda_\beta},\frac{\rho_2}{\lambda_\beta})+\frac{\rho_1}{\lambda_\beta}\left[-\frac{\xi_1}{\lambda_\beta}\Psi_2(\frac{\xi_1}{\lambda_\beta},\frac{\xi_2}{\lambda_\beta})+\frac{\zeta_1}{\lambda_\beta}\Psi_2(\frac{\zeta_1}{\lambda_\beta},\frac{\zeta_2}{\lambda_\beta})\right]\right.$$

$$\left.+\frac{\rho_2}{\lambda_\beta}\left[\frac{\xi_2}{\lambda_\beta}\Psi_3(\frac{\xi_1}{\lambda_\beta},\frac{\xi_2}{\lambda_\beta})-\frac{\zeta_2}{\lambda_\beta}\Psi_3(\frac{\zeta_1}{\lambda_\beta},\frac{\zeta_2}{\lambda_\beta})\right]\right\},\tag{20c}$$

where $\rho_1=\xi_1-\zeta_1$, $\rho_2=\xi_2-\zeta_2$, and the description of functions $\Theta_1$ through $\Theta_3$, or $\Psi_1$

through $\Psi_3$, can be found in Appendix B. Detailed derivations of Eq. (18) to Eq. (20)

can be found in Appendix B.

In the case of statistically nonhomogeneous random fields, the structure of variability can be characterized by considering the semivariogram of a random field. If the semivariogram depends only on the separation, the random field is said to have stationary increments. The semivariogram in Eq. (20) clearly depends on the spatial location, which means that the processes of depth-averaged hydraulic head are nonstationary.

[Figure]

**Figure 1.** The stationary parts of the semivariogram of the head field, reflecting the effect of variation in the hydraulic conductivity fields, as a function of the separation distance in the mean flow direction, where $G_y$ is the sum of the first three terms on the right-hand side of Eq. (20b).

Figure 1 shows graphically the behavior of the stationary parts of the semivariogram (namely, the sum of the first three terms on the right-hand side of Eq.

(20b)) as a function of the separation distance in the $x_1$-direction (mean flow direction).

The semivariogram of the head field, reflecting the effect of variation in the hydraulic conductivity fields, shows an unlimited increase, as shown in Fig. 1. The unbounded head semivariogram suggests that there is no head covariance function (or the hydraulic head field with infinite variance). When taking samples from a field, one obtains a histogram from which a certain value of the variance can always be calculated. However, for many phenomena, the experimental variance is actually a function of the field. In particular, it increases as the field increases, i.e., many phenomena have an almost unlimited capacity of dispersion and cannot be adequately described by ascribing to them a finite a priori variance. In this case, the use of the semivariogram is an appropriate way to measure the variability of the variation.

Similar conclusions can be drawn from Fig. 2, a graphical representation of the stationary parts of the semivariogram of the head field in Eq. (20c) in the mean flow direction, which reflects the effect of the variation of the aquifer thickness fields.

At this point, the covariance function for the depth-averaged velocity process in

Eq. (13) can now be determined in conjunction with Eqs. (16), and (18)-(20). For example, the covariance of flow velocity for the separation along the mean flow direction is explicitly determined as follows:

$<v_1(\xi_1,\xi_2)v_1(\zeta_1,\zeta_2=\xi_2)>=<v_{y_1}(\xi_1,\xi_2)v_{y_1}(\zeta_1,\xi_2)>+<v_{\beta_1}(\xi_1,\xi_2)v_{\beta_1}(\zeta_1,\xi_2)>,$    (21a)

where

$$\quad \frac{<v_{y_1}(\xi_1,\xi_2)v_{y_1}(\zeta_1,\xi_2)>}{V^2}=\sigma_y^2\left\{\frac{3}{8}+\exp(-\frac{\rho}{\lambda_y})-\left[2\Xi_1(\frac{\rho_1}{\lambda_y},0)-\Xi_1(\frac{\xi_1}{\lambda_y},\frac{\xi_2}{\lambda_y})-\Xi_1(\frac{\zeta_1}{\lambda_y},\frac{\xi_2}{\lambda_y})\right]\right.$$

$$\qquad\qquad \left.+\left[\Xi_2(\frac{\rho_1}{\lambda_y},0)-\Xi_2(\frac{\xi_1}{\lambda_y},\frac{\xi_2}{\lambda_y})-\Xi_2(\frac{\zeta_1}{\lambda_y},\frac{\xi_2}{\lambda_y})\right]\right\}, \tag{21b}$$

$$\quad \frac{<v_{\beta_1}(\xi_1,\xi_2)v_{\beta_1}(\zeta_1,\xi_2)>}{V^2}=\sigma_\beta^2\left\{\frac{3}{2}+\exp(-\frac{\rho}{\lambda_\beta})-2\left[2\Xi_1(\frac{\rho_1}{\lambda_\beta},0)-\Xi_1(\frac{\xi_1}{\lambda_\beta},\frac{\xi_2}{\lambda_\beta})-\Xi_1(\frac{\zeta_1}{\lambda_\beta},\frac{\xi_2}{\lambda_\beta})\right]\right.$$

$$\qquad\qquad \left.+4\left[\Xi_2(\frac{\rho_1}{\lambda_\beta},0)-\Xi_2(\frac{\xi_1}{\lambda_\beta},\frac{\xi_2}{\lambda_\beta})-\Xi_2(\frac{\zeta_1}{\lambda_\beta},\frac{\xi_2}{\lambda_\beta})\right]\right\}, \tag{21c}$$

$\rho=(\rho_1^2+\rho_2^2)^{1/2}$ and expressions for $\Xi_1$ and $\Xi_2$ are given, respectively, in the Appendix C.

This should be used to compute the variance of solute displacement in the mean flow direction. The nonstationarity of the velocity covariance in Eq. (21) is evident in the dependence on spatial location, which is caused by nonstationarity in the hydraulic head processes.

[Figure]

**Figure 2.** The stationary parts of the semivariogram of the head field, reflecting the effect of the variation of the aquifer thickness fields, as a function of the separation distance in the mean flow direction, where $G_\beta$ is the sum of the first three terms on the right-hand side of Eq. (21c).

In the limit of $\zeta_1 \to \xi_1$, Eq. (21) approaches to the velocity variances in the mean flow direction as

$\sigma_v^2 = \sigma_{v_y}^2(\xi_1, \xi_2) + \sigma_{v_\beta}^2(\xi_1, \xi_2),$                    (22a)

where

$\dfrac{\sigma_{v_y}^2}{V^2 \sigma_y^2} = \dfrac{1}{4}\dfrac{1}{\xi^8}\Delta_1(\dfrac{\xi_1}{\lambda_y}, \dfrac{\xi_2}{\lambda_y}) + \dfrac{1}{4}\dfrac{1}{\xi^9}\exp[-\dfrac{\xi}{\lambda_y}]\Delta_2(\dfrac{\xi_1}{\lambda_y}, \dfrac{\xi_2}{\lambda_y}),$                    (22b)

$\dfrac{\sigma_{v_\beta}^2}{V^2 \sigma_\beta^2} = \dfrac{1}{2}\dfrac{1}{\xi^8}\Delta_3(\dfrac{\xi_1}{\lambda_\beta}, \dfrac{\xi_2}{\lambda_y}) + \dfrac{1}{2}\dfrac{1}{\xi^9}\exp[-\dfrac{\xi}{\lambda_\beta}]\Delta_4(\dfrac{\xi_1}{\lambda_\beta}, \dfrac{\xi_2}{\lambda_\beta}),$                    (22c)

$\xi = (\xi_1^2 + \xi_2^2)^{1/2}$ and expressions for $\Delta_1$-$\Delta_4$ are given, respectively, in the Appendix D.

From Eq. (22), it can be seen that the variance of the flow velocity is positively correlated with the variances of the log-hydraulic conductivity and log-aquifer thickness. This means that the variability of the flow velocity field increases with the variability of the hydraulic conductivity and aquifer thickness fields.

**4.2  Variance of the solute displacement in the mean flow direction**

**4.2.1      Nonstationary flow fields**

Substituting Eq. (21) into Eq. (15) and integrating it yields the following expression for the variance of longitudinal solute displacement as

$X_{11}(t) = X_{11_y}(t) + X_{11_\beta}(t)$,      (23a)

where

$\dfrac{X_{11_y}(t)}{\sigma_y^2 \lambda_y^2} = \dfrac{5}{2} - 3\gamma - \dfrac{9}{\Gamma^2} + 2\Gamma + \dfrac{3}{8}\
[revised manuscript text omitted]
. Note that in developing Eq. (A2), it is assumed that the contaminant plume in confined aquifers is well mixed over depth, so that variations around the depth-averaged concentration are relatively small (Holly, 1975). Then the average of the product of concentration and velocity fluctuations can be assumed to be absorbed in the gradient transport terms in Eq. (A2)

Starting from the identity,

$$\frac{\tilde{D}_i}{n}B\frac{\partial \tilde{c}}{\partial x_i} = \frac{\partial}{\partial x_i}[\frac{\tilde{D}_i}{n}B\tilde{c}] - \tilde{c}\frac{\partial}{\partial x_i}[\frac{\tilde{D}_i}{n}B]$$

$$= \frac{\partial}{\partial x_i}[\frac{\tilde{D}_i}{n}B\tilde{c}] - B\tilde{c}\frac{1}{n}\frac{\partial \tilde{D}_i}{\partial x_i} - \frac{\tilde{D}_i}{n}B\tilde{c}\frac{\partial \ln B}{\partial x_i} \qquad i = 1,2, \qquad (A3)$$

Eq. (A2) can be rewritten as follows:

$$\frac{\partial}{\partial t}[B\tilde{c}] = \frac{\partial^2}{\partial x_i^2}[\frac{\tilde{D}_i}{n}B\tilde{c}] - \frac{\partial}{\partial x_i}[(\frac{1}{n}\frac{\partial \tilde{D}_i}{\partial x_i} + \frac{\tilde{D}_i}{n}\frac{\partial \ln B}{\partial x_i} + \frac{\tilde{q}_i}{n})B\tilde{c}] \qquad i = 1,2, \tag{A4}$$

which corresponds to the form of the Fokker-Planck equation (e.g., Risken, 1989).

The concentration field associated with the solute particle can be written as (Fischer et al., 1979; Dagan, 1989)

$$B\tilde{c} = \frac{M}{n}f(\boldsymbol{x};t,\boldsymbol{a},t_0), \tag{A5}$$

where $M$ is the solute mass, $f(\boldsymbol{x};t,\boldsymbol{a},t_0)$ stands for the probability density function of the particle displacement which originates at $\boldsymbol{x} = \boldsymbol{a}$ for $t = t_0$. Substituting Eq. (A5) into Eq.

(A4) gives

$$\frac{\partial}{\partial t}f(\boldsymbol{x};t) = \frac{\partial^2}{\partial x_i^2}[\frac{\tilde{D}_i}{n}f(\boldsymbol{x};t)] - \frac{\partial}{\partial x_i}[(\frac{1}{n}\frac{\partial \tilde{D}_i}{\partial x_i} + \frac{\tilde{D}_i}{n}\frac{\partial \ln B}{\partial x_i} + \frac{\tilde{q}_i}{n})f(\boldsymbol{x};t)] \qquad i = 1,2, \tag{A6}$$

which is known as the Fokker-Planck equation. Moreover, it can be shown that the stochastic differential equation for the evolution of stochastic process (e.g., Van

Kampen, 1992; Jing et al., 2019)

$$\frac{dX_i}{dt} = \mu_i(X(t)) + \sigma_i(X(t))\frac{dW}{dt} \qquad i = 1,2, \tag{A7}$$

where $X(= (X_1, X_2))$ is the displacement, $\mu_i$ is the drift coefficient, $\sigma_i$ is the diffusion coefficient, and $W$ denotes a Wiener process, is equivalent to the Fokker-Planck equation (A6) such that

$$\mu_i = \frac{1}{n}\frac{\partial}{\partial x_i}\tilde{D}_i(X) + \frac{1}{n}\tilde{D}_i(X)\frac{\partial}{\partial x_i}\ln B(X) + \frac{1}{n}\tilde{q}_i(X) \qquad i = 1,2, \tag{A8a}$$

$$\sigma_i^2 = \frac{2}{n}\tilde{D}_i(X) \qquad i = 1,2, \tag{A8b}$$

Using Eq. (A8), Eq. (A7) leads to Eq. (2).

**Appendix B: Derivations of Eq. (18) to Eq. (20)**

Due to the property of the linearity of the driving forces, Eq. (10b) can alternatively be divided into two parts as

$$\frac{\partial^2 h_y}{\partial x_i^2} = J \frac{\partial y}{\partial x_1} \qquad i=1,2, \tag{B1a}$$

$$\frac{\partial^2 h_\beta}{\partial x_i^2} = 2J \frac{\partial \beta}{\partial x_1} \qquad i=1,2, \tag{B1b}$$

where $h = h_y + h_\beta$. Matheron (1973) shows that if the random input process of the

Poission equation is second-order stationary, then the Poission equation has a first-order intrinsic random function (1-IRF) as its solution. Since the processes $y$ and

$\beta$ are second-order stationary, it can be shown that the derivatives of the processes $y$

and $\beta$ with respect to $x_1$ are also stationary. This means that Eq. (B1) has a 1-IRF

solution for $h_y$ and $h_\beta$ which admits the Fourier-Stieltjes representation as follows:

$$h_y(x_1, x_2) = J \int_{-\infty}^{\infty} \int_{-\infty}^{\infty} iR_1 \frac{1 - \exp[i(R_1 x_1 + R_2 x_2)] + i(R_1 x_1 + R_2 x_2)}{R_1^2 + R_2^2} dZ_y(R_1, R_2), \tag{B2a}$$

$$h_\beta(x_1, x_2) = 2J \int_{-\infty}^{\infty} \int_{-\infty}^{\infty} iR_1 \frac{1 - \exp[i(R_1 x_1 + R_2 x_2)] + i(R_1 x_1 + R_2 x_2)}{R_1^2 + R_2^2} dZ_\beta(R_1, R_2). \tag{B2b}$$

where $R_1$ and $R_2$ are the components of the wave number vector $\mathbf{R}$ (= $(R_1, R_2)$), and $Z_y$

and $Z_\beta$ are complex-valued distributions with uncorrelated increments on wave number space. Note that a 1-IRF is the second integral of a zero-mean spatial random function (Chile`s and Delfiner, 1999).

The stationarity of the ln$K$ process allows the Fourier-Stieltjes representations (e.g., Lumley and Panofsky, 1964)

$$\quad y(x_1, x_2) = \int_{-\infty}^{\infty}\int_{-\infty}^{\infty} \exp[i(R_1 x_1 + R_2 x_2)]dZ_y(R_1, R_2). \tag{B3}$$

Using this and Eqs. (B2a) and (24a), the covariance of ln$K$ process with the head process $C_{yh}$ in Eq. (12) is given as

$$\quad C_{yh_y}(\xi,\zeta) = \langle y(\xi)h_y(\zeta) \rangle$$

$$\quad = -J\int_{-\infty}^{\infty}\int_{-\infty}^{\infty} i\frac{R_1}{R_1^2+R_2^2}\exp[i(R_1\xi_1+R_2\xi_2)]\left\{1-\exp[-i(R_1\zeta_1+R_2\zeta_2)]-i(R_1\zeta_1+R_2\zeta_2)\right\}$$

$$\quad \times \frac{\sigma_y^2}{2\pi}\frac{\lambda_y^2}{[1+\lambda_y^2(R_1^2+R_2^2)]^{3/2}}dR_1 dR_2$$

$$\quad = \sigma_y^2\lambda_y J\left[\Theta_1(\frac{\xi_1}{\lambda_y},\frac{\xi_2}{\lambda_y})-\frac{\zeta_1}{\lambda_y}\Theta_2(\frac{\xi_1}{\lambda_y},\frac{\xi_2}{\lambda_y})+\frac{\zeta_2}{\lambda_y}\Theta_3(\frac{\xi_1}{\lambda_y},\frac{\xi_2}{\lambda_y})-\Theta_1(\frac{\rho_1}{\lambda_y},\frac{\rho_2}{\lambda_y})\right], \tag{B4}$$

where $\rho_1 = \xi_1 - \zeta_1$, $\rho_2 = \xi_2 - \zeta_2$, and

$$\quad \Theta_1(a,b) = \frac{a}{r}[1-e^{-r}(1+r)], \tag{B5a}$$

$$\quad \Theta_2(a,b) = -2\frac{a^2}{r^4}+\frac{1}{r^2}+e^{-r}[a^2(\frac{2}{r^4}+\frac{2}{r^3}+\frac{1}{r^2})-\frac{1}{r^2}-\frac{1}{r}], \tag{B5b}$$

$$\quad \Theta_3(a,b) = ab[\frac{2}{r^4}-e^{-r}(\frac{2}{r^4}+\frac{2}{r^3}+\frac{1}{r^2})], \tag{B5c}$$

$r^2 = a^2+b^2$.

Similarly, the closed-form expression for the covariance of $\ln B$ process with the head process $C_{\beta h_\beta}$ in Eq. (12) can be obtained using Eqs. (B2b), (24b), and the

Fourier-Stieltjes representations for the stationary $\ln B$ process

$$\beta(x_1, x_2) = \int_{-\infty}^{\infty}\int_{-\infty}^{\infty} \exp[i(R_1 x_1 + R_2 x_2)]dZ_\beta(R_1, R_2),\tag{B6}$$

which is in the form

$C_{\beta h_\beta}(\xi,\zeta) = <\beta(\xi)h_\beta(\zeta)>$

$$= 2\sigma_y^2\lambda_\beta J\left[\Theta_1(\frac{\xi_1}{\lambda_\beta},\frac{\xi_2}{\lambda_\beta}) - \frac{\zeta_1}{\lambda_\beta}\Theta_2(\frac{\xi_1}{\lambda_\beta},\frac{\xi_2}{\lambda_\beta}) + \frac{\zeta_2}{\lambda_\beta}\Theta_3(\frac{\xi_1}{\lambda_\beta},\frac{\xi_2}{\lambda_\beta}) - \Theta_1(\frac{\rho_1}{\lambda_\beta},\frac{\rho_2}{\lambda_\beta})\right].\tag{B7}$$

Substituting Eq. (B2) into Eq. (13), it is found that the semivariogram of the head process has the following form

$$\gamma_{h_y}(\xi,\zeta) = \frac{1}{2}\sigma_y^2\lambda_y^2 J^2\left\{\frac{3}{8}\frac{\rho_1^2}{\lambda_y^2} + \frac{1}{8}\frac{\rho_2^2}{\lambda_y^2} + \Psi_1(\frac{\rho_1}{\lambda_y},\frac{\rho_2}{\lambda_y}) + \frac{\rho_1}{\lambda_y}[-\frac{\xi_1}{\lambda_y}\Psi_2(\frac{\xi_1}{\lambda_y},\frac{\xi_2}{\lambda_y}) + \frac{\zeta_1}{\lambda_y}\Psi_2(\frac{\zeta_1}{\lambda_y},\frac{\zeta_2}{\lambda_y})]\right.$$

$$\left. + \frac{\rho_2}{\lambda_y}[\frac{\xi_2}{\lambda_y}\Psi_3(\frac{\xi_1}{\lambda_y},\frac{\xi_2}{\lambda_y}) - \frac{\zeta_2}{\lambda_y}\Psi_3(\frac{\zeta_1}{\lambda_y},\frac{\zeta_2}{\lambda_y})]\right\},\tag{B8a}$$

$$\gamma_{h_\beta}(\xi,\zeta) = 2\sigma_\beta^2\lambda_\beta^2 J^2\left\{\frac{3}{8}\frac{\rho_1^2}{\lambda_\beta^2} + \frac{1}{8}\frac{\rho_2^2}{\lambda_\beta^2} + \Psi_1(\frac{\rho_1}{\lambda_\beta},\frac{\rho_2}{\lambda_\beta}) + \frac{\rho_1}{\lambda_\beta}[-\frac{\xi_1}{\lambda_\beta}\Psi_2(\frac{\xi_1}{\lambda_\beta},\frac{\xi_2}{\lambda_\beta}) + \frac{\zeta_1}{\lambda_\beta}\Psi_2(\frac{\zeta_1}{\lambda_\beta},\frac{\zeta_2}{\lambda_\beta})]\right.$$

$$\left. + \frac{\rho_2}{\lambda_\beta}[\frac{\xi_2}{\lambda_\beta}\Psi_3(\frac{\xi_1}{\lambda_\beta},\frac{\xi_2}{\lambda_\beta}) - \frac{\zeta_2}{\lambda_\beta}\Psi_3(\frac{\zeta_1}{\lambda_\beta},\frac{\zeta_2}{\lambda_\beta})]\right\},\tag{B8b}$$

where

$$\Psi_1(a,b) = \frac{a^2 - b^2}{r^2}[\frac{1}{2} + \frac{e^{-r}(r^2 + 3r + 3) - 3}{r^2}] - Ei(r) + \ln(r) + e^{-r} - 1 + \gamma,\tag{B9a}$$

$$\Psi_2(a,b) = \frac{1}{r^6}(a^4 + 6a^2 + 4a^2 b^2 + 3b^4 - 18b^2) + e^{-r}[-2\frac{a^6}{r^7} - a^4(\frac{6}{r^7} + \frac{4}{r^6}) - 6\frac{a^2}{r^6} - 4\frac{a^4 b^2}{r^7}$$

$$+ 2a^2 b^2(\frac{6}{r^7} + \frac{1}{r^6}) - 2\frac{a^2 b^4}{r^7} + 6b^4(\frac{3}{r^7} + \frac{1}{r^6}) + 18\frac{b^2}{r^6}],\tag{B9b}$$

$$\Psi_3(a,b) = \frac{1}{r^6}(a^4 - 18a^2 - b^4 + 6b^2) + e^{-r}[2\frac{a^6}{r^7} + 2a^4(\frac{9}{r^7} + \frac{4}{r^6}) + 18\frac{a^2}{r^6} + 4\frac{a^4 b^2}{r^7}$$

$$+2\frac{a^2b^4}{r^7}+6a^2b^2(\frac{2}{r^7}+\frac{1}{r^6})-2b^4(\frac{3}{r^7}+\frac{1}{r^6})-6\frac{b^2}{r^6}],$$
(B9c)

[revised manuscript text omitted]